# Time2Image: A Unified Adaptive Image Representation Framework for Time Series Classification

## Abstract

Time Series Classification (TSC) is a crucial and challenging task that holds significant importance across various domains, of which one of the kernel ingredients is to construct a suitable time series representation for better feature capture. However, extracting informative and robust time series representation with good generalization potential is still a challenging problem. To address this issue, we propose Time2Image, a novel image-based representation framework for TSC. At the heart of our framework is a proposed Adaptive Time Series Gaussian Mapping (ATSGM) module for robust time series encoding in 2D image structure, based on which we employ Vision Transformer (ViT) for subsequent classification tasks considering its prominent long-dependency modeling capability. Experiments were conducted on all 158 public time series datasets from UCR/UEA covering diverse domains, among which our method achieves top 1 performance in 86 datasets compared with existing State-Of-The-Art (SOTA) deep learning-based methods. In addition, our framework flexibly allows handling both univariate and multivariate time series with unequal length across different domains and takes inherent advantage of generalization ability due to our proposed ATSGM representation method. The source code will be publicly available soon.

## 1 Introduction

Time series classification (TSC) is recognized as a classic but challenging task in data mining (Esling & Agon, 2012), which aims to assign predefined labels to chronologically arranged data of both Univariate Time Series (UTS) and Multivariate Time Series (MTS) according to the number of channels of the sample. It can be widely applied across diverse fields in finance(Xiu et al., 2021; Chao et al., 2019), healthcare(Chambon et al., 2018), transportation(Gupta et al., 2020), etc. Over the past few years, TSC algorithms can be mainly concluded into 3 categories:(i) Traditional machine learning models(Formisano et al., 2008; Bagnall et al., 2017) use various feature extraction techniques for statistic(Lin et al., 2012; Li et al., 2018), frequency(Baydogan et al., 2013), sequence(Chen et al., 2021) or shapelet(Ye & Keogh, 2009; Grabocka et al., 2014) feature capturing combined with traditional classification methods(Xue et al., 2019) like SVM, KNN, etc. (ii) Deep learning models(Chen & Shi, 2019; Ruiz et al., 2021) have automatic feature learning ability through neural network models to achieve more substantial expressive power compared with traditional methods. Typical algorithms for sequence modeling ability including RNN, LSTM, especially Transformer-related models based on attention mechanism on long-term dependencies capturing. (iii) Ensemble models(Lines et al.) integrate the results by combining multiple base classifiers to improve classification performance. However, existing algorithms are only suitable for either UTS or MTS with heavy feature engineering and hyperparameter tuning, which brings subjectivity to the model.

Unlike the above models which extract time series representation based on original time series data, in recent years, increasing attention has been focused on transformation-based time series representation(Bagnall et al., 2012). These methods model time series data with specific data structure for informative feature extraction, among which time series image representation has become one of the active areas in recent years with the rapid development and achievements of image classification algorithms in computer vision(Chen & Shi, 2019). The motivation behind image representation is to convert time series into images to reformat the data for effective pattern detection to strengthen

the expressive power of the data by leveraging experience in image feature extraction. However, current image representation methods suffer from poor generalization, which can be reflected in two aspects: from the data perspective, current approaches are only effective in specific time series datasets or in certain domains; from the model perspective, existing image representation methods cannot be applied to both UTS and MTS. Even though some models can be adopted on MTS, many of them cannot be used when the lengths of time series are inconsistent. Therefore, our goal of this work is to propose a novel time series image representation framework that not only has a better comprehensive performance compared with existing deep learning SOTA algorithms but also has the inherent generalization ability to both UTS and MTS with inconsistent length.

In this paper, we proposed a unified adaptive image representation framework for time series classification called Time2Image. In our framework, Adaptive Time Series Gaussian Mapping (ATSGM) is first introduced to convert time series into an image consisting a collection of mixed Gaussian images where the image number equals the length of the time series data. Moreover, each mixed Gaussian image is jointly constructed based on a specific two-dimensional Gaussian distribution and the values of the time series data at a certain time point. By converting the projection of the time series data into an 'equal circle in a square' problem, the optimal value of the specific Gaussian distribution parameters and the position of each channel in the image can be obtained given channel number and image size. After that, the time series classification is converted into an image classification problem, and the vision transformer algorithm is adopted with the help of its long-term dependency-capturing ability. This design enables spatial structure construction of time series through image representations and can be generalized to both UTS and MTS with unequal lengths. Overall, the contributions can be summarized as follows:

- Adaptive Time Series Gaussian Mapping (ATSGM) module is proposed for robust time series encoding in 2D images, which can be generalized to both UTS and MTS.

- The vision transformer adopted in Time2Image is the first attempt at a time series classification task.

- We validate the effectiveness of our approach based on all 158 public datasets from UCR/UEA. Experimental results show that our approach achieves notably superior performance compared with SOTA baselines.

## 2 RELATED WORK

### 2.1 TIME SERIES TRANSFORMATION METHODS

With the accumulation of time series data in various domains, transforming time series into alternative representations has become crucial for advanced analysis tasks as a way to improve the expressive power of original data(Lacasa et al., 2015)(Meintjes et al.). Graph-based transformation method is a flexible framework to capture complex interrelationships and dependencies within a time series(Cheng et al., 2020). Techniques such as Visibility graph(Xiu et al., 2022), Recurrence network(Donges et al., 2012), and Transition network (Makaram et al., 2021) are available for time series modeling. Under this framework, graph theory and network science can be adopted for further tasks but constructing a graph is computationally expensive, especially for long time series data. Moreover, symbolic sequence representation aims to simplify continuous time series data into discrete symbols based on predefined rules. A Method like Symbolic Aggregation approXimation (SAX)(Senin & Malinchik, 2013) is proposed for representation, which allows the utilization of symbolic analysis, but it will inevitably lose detailed information and the selection of the parameters is subjective. In the meantime, numerical transformation includes Fourier Transform(Zhao et al., 2017), Wavelet Transform(Chaovalit et al., 2011), etc. endeavor to execute mathematical operations for spectral component capturing or features from different scales, but the estimation and selection of suitable transformation functions can also be subjective.

In addition to the above methods, image-based representation has gained popularity in recent years with the development of computer vision. Existing image-encoder methods (Li et al., 2021; Wang & Oates, 2015; Chen & Shi, 2019) for time series include Gramian Angular Field (GAF), Markov Transition Field (MTF), Recurrence Plots (RP), etc. Phase relationships, recurrence patterns, and frequency-related features can be captured through current techniques. Since there is a significant

gap between the existing time series image representation method for classification and the SOTA models on the TSC task, we propose a new time series image representation method in this paper.

## 2.2 IMAGE CLASSIFICATION

When it comes to image classification, various deep learning architectures have emerged as state-of-the-art models for image classification. Existing architectures can be concluded into 2 categories: Convolutional Neural Networks(CNNs)(Esling & Agon, 2012; Li et al., 2021) based models and Transformer based models(Dosovitskiy et al., 2021). CNNs have revolutionized this field, achieving remarkable results by effectively capturing local spatial dependencies through convolutional layers and hierarchical features via pooling and stacking operations, of which ResNet(He et al., 2016) is a typical model of CNN-based models. More recently, attention mechanisms have gained attention in image classification research. After that, the emergence of ViT from Google proposed in 2021(Dosovitskiy et al., 2021) indicates that the transformer-based models have officially entered the field of image classification. However, ViT has never been applied to TSC tasks before. Since it has a good long-dependence modeling capability, it should have great potential to be applied to temporal data. In this work, by converting time series into image, we transform the time series classification into image classification and utilize vision transformer for further tasks.

## 3 PRELIMINARY

Let $\chi_N = \left\{X_D^N\right\}_{d=1}^D$ be the $N^{th}$ multivariate time series data with the dimension of $D$. $X_D \in \mathbb{R}^{D \times T}$ refers to the $D^{th}$ channel of time series and $X_D = \{x_{d_1,1}, x_{d_2,2} ....., x_{d,t}\}$. For $\forall \chi$, $D$ and $T$ represent the channel and the length of the time series, respectively. Let $Y_N \in \mathbb{N}^{\mathbb{K}}$ be the corresponding label of the $N^{th}$ sample of the time series, where $K$ indicates the number of classes. All channels in $X_N$ share the same label $Y$. We choose the definition of multivariate time series as the general definition of both univariate and multivariate time series data since univariate time series can be regarded as the special case of multivariate ones when $D = 1$. In this study, we focus on time series classification by transforming the original time series into an image (Time2Image). Our Time2Image consists of two stages: Adaptive Time Series Gaussian Mapping (ATSGM) for image representation and classification.

**Definition 1 Patch.** A patch refers to a small rectangular or square region extracted from the input image, which can be mathematically represented as a matrix or a vector. It is a fundamental unit in computer vision, which plays a vital role in local feature encoding and analysis. In addition, the shape and size of the patch are adaptable based on the application and models we adopt, of which smaller patches reflect fine-grained details while larger patches encompass a broader context. In this work, the patch $P_t$ is defined as the image representation of the time series at time $t$, which is a $16 \times 16$ matrix since the classification method we adopt is ViT-B/16.

**Definition 2 Sub-patch.** A sub-patch is defined as the subsection of the patch in definition 1. As for MTS, the image representation of the time series in one channel is a sub-patch. Therefore, the number of sub-patch of a MTS sample equals the number of channels. Therefore, UTS can be regarded as a special case of MTS, of which the sub-patch and patch are the same.

## 4 TIME2IMAGE FRAMEWORK

In this section, a novel time series image representation framework is introduced for time series modeling. We name the proposed framework as Time2Image, which transforms time series into an image. The framework can be seen in Figure 1, from which we use D=6 as an example.

### 4.1 DATA PREPROCESSING

Data preprocessing plays a critical role in preparing the time series data for classification tasks. In this framework, the data preprocessing involves two techniques, which are standardization and resizing. For time series data of each channel in MTS, standardization is first conducted separately to align data to a common scale and distribution so as to ensure different time series from different

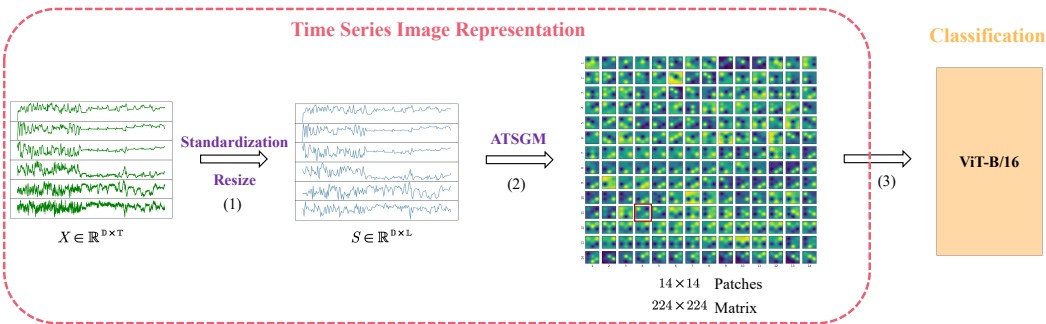

Figure 1: Time2Image Framework (1)Pre-processing: use standardization and resize to let MTS to equal-length MTS and L=196 (2) ATSGM:Gaussian mapping to model time series into a mixed Gaussian distribution as image representation (3) Use the image generated from ATSGM for image classification task

channels are comparable.

$$S_{D,T}^N = \frac{X_{D,T}^N - \mu_{X_D^N}}{\sigma_{X_D^N}} \tag{1}$$

where $\mu$ and $\sigma$ are the mean and standard deviation of the time series, respectively. After that, cubic interpolation is adopted for each channel to deal with varying sequence lengths within the time series to create a consistent representation. Since the estimation process is determined through smooth cubic polynomial, it provides more accurate results, especially for complex time series data with nonlinear variations compared to simpler interpolation methods such as linear interpolation and quadratic interpolation.

## 4.2 ADAPTIVE TIME SERIES GAUSSIAN MAPPING (ATSGM)

ATSGM is a crucial component of our proposed framework for time series image representation, which addresses the challenge of extracting informative and robust representations from time series data with the goal of achieving better feature capture. Our goal is to obtain an image representation of the corresponding values of all channels at a certain time. The overall process of ATSGM can be shown in Figure 2, which involves transforming the time series at a certain time with different channels into a sequence of mixed Gaussian distributions ordered by sequence. These distributions are then used to create a sub-patch representation, where the mean and standard deviation of the Gaussian distribution correspond to the specific value through mathematical derivation based on the number of channels of MTS, which is illustrated in Section 4.2.1. The summation of the sub-patch representation is conducted and the patch representation is reached for time series at time $t$. All obtained patches are arranged in chronological order into $16 \times 16$ patches as the image representation of MTS as the input for the image classification algorithm. The intuition of the ATSGM is to preserve the statistical properties of time series through Gaussian distributions and obtain a smooth two-dimensional representation. The following subsection will give a detailed description of the method.

### 4.2.1 TIME SERIES IMAGE REPRESENTATION

Existing research on image representation mainly considers the relative value by simply getting the difference between different time steps, but here we consider a two-dimensional Gaussian distribution in which the covariance matrix is zero in default and the two standard deviations are equal. Therefore, the projection of this Gaussian distribution is a circle in the plane, where the radius of the circle equals the standard deviation of the Gaussian distribution. Moreover, the mean $\mu_x$ and $\mu_y$ can be regarded as the coordination of the center of the circle. After that, the projection value of 2D Gaussian distribution is constructed as the sub-patch matrix, of which the length and size of the patch are predefined as a $16 \times 16$ matrix with the length of each patch equals 6, and the value is defined in a range [-3,3]. The value of the fundamental Gaussian distribution for the sub-patch matrix can be obtained through the following equation.

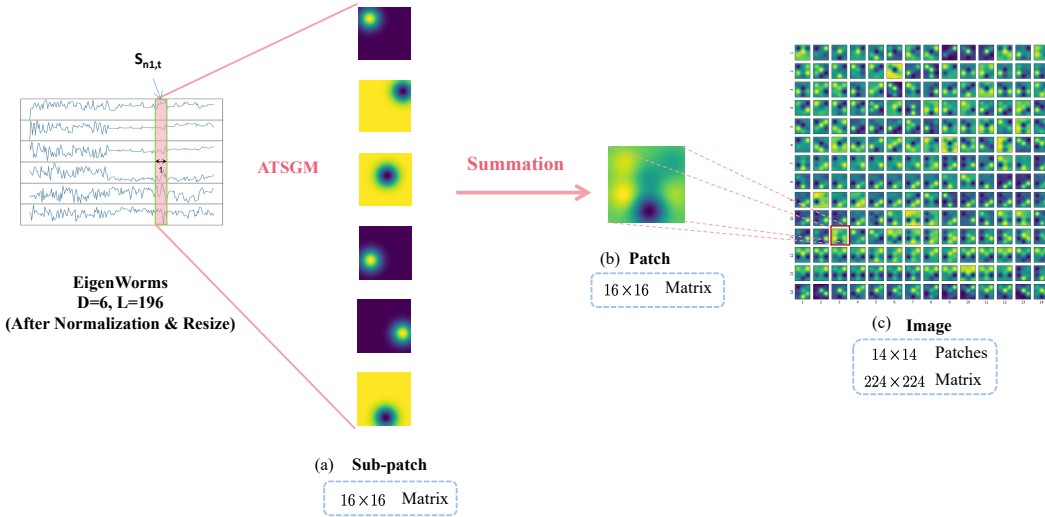

Figure 2: Time series image representation (a)Sub-patch: For pre-processed multivariate time series data, use ATSGM to get gaussian mapping of each channel at a certain time stamp (b)Patch: Do summation of sub-patch from all channels at a certain time stamp to get the patch at a certain time stamp (c) Image: Patches combined with position encoding connected in chronological order to get the final image

$$f(x, y) = \frac{1}{2\pi\sigma^2} \exp\left[-\frac{1}{2}\left(\frac{(x-\mu_x)^2}{\sigma^2} + \frac{(y-\mu_y)^2}{\sigma^2} - \frac{2(x-\mu_x)(y-\mu_y)}{\sigma^2}\right)\right] \quad (2)$$

Where $f(x, y)$ stands for the matrix value at $(x, y)$, $\mu_x$,$\mu_y$ and $\sigma$ refers to the mean and standard deviation of the distribution, respectively. Since the projection of 2D Gaussian distribution is a circle in the plane, the relationship between the area of the circle and the standard deviation of Gaussian distribution can be derived as:

$$S_{circle} = \pi R_d^2 = \pi(\sigma)^2 \quad (3)$$

where $R_D$ is the radius of the circle in a $D$-channel times series, from which we can obtain that the radius equal to the standard deviation of 2D Gaussian distribution. Here adaptive from ATSGM refers to the adjustable of the standard deviation, that is to say, we can get the representation with different information by setting different values of standard deviation. The smaller the standard deviation, the more information is captured from Gaussian mapping. According to '$3sigma$' principle, we can derive the corresponding relationship between $R_D$ and the value of standard deviation as follows:

- When $\sigma = R_D$, about 68% of the information can be represented within the circle.
- When $\sigma = R_D/2$, about 95% of the information can be represented within the circle.
- When $\sigma = R_D/3$, about 99% of the information can be represented within the circle.

Therefore, the projection value $V_{d,t}$ of channel d at time t in the coordination of sub-patch matrix (x,y) is defined as:

$$V_{d,t}(x, y) = f(x, y) \times S_{d,t} \quad (4)$$

Where $S_{d,t}$ is the preprocessed time series value at time $t$. After the calculation of all data points, the characteristic of the randomness of the time series data point for each channel can be captured. Here we use the Gaussian distribution to describe the randomness of the value, adjust the range and strength of the Gaussian distribution by multiplying the normalized specific value of the time series data, and use the adjusted distribution of each dimension as the binary value under timestep dimensional representation to improve the stability and robustness of the method.

### 4.2.2 SUB-PATCH POSITION DETERMINATION

From the construction process of ATSGM above, we can conclude that for UTS, the optimal time series image representation can be obtained when the center of the projected circle is located at the center of the sub-patch and the diameter equals the length of the sub-patch. However, when it comes to MTS, the projection position needs to be determined first for each channel. Since the projection of 2D Gaussian distribution is a circle in the plane, we can regard it as a packing problem, which is to find the best packings of equal circles in a square. In fact, the "equal circle in a square" is a mathematical puzzle that involves finding the largest possible circle that can fit inside a given square, such that the circle's diameter is equal to the side length of the square. In other words, the goal is to determine the maximum-sized circle that can be inscribed within the square. Website[1] shows the best-known packings of equal circles in a square from N=1 to 10000, including the optimal radius ($r_d$) and the corresponding coordinates ($c_d$) of each circle given $N$ when the length of the square is 1. In our work, N equals the number of channels in MTS. Therefore, the radius and coordinates can be obtained as:

$$R_d = r_d \times 6 \tag{5}$$

$$C_d = (c_{dx} \times 6, c_{dy} \times 6) \tag{6}$$

After finding out the optimal radius of the patch, the optimal parameters of Gaussian distribution can be determined, of which the $\mu_x$ and $\mu_y$ equal the coordinates from Equation 6, and the standard deviation can also be obtained through Equation 3. After the determination of the parameters, the distribution of Gaussian will be finally determined for each sub-patch representation. The patch representation of time step $t$ is achieved by summing all sub-patch representations at a certain time step, which is shown in Equation 6. The image representation is the arrangement of different Patches ordered by sequence.

$$P_t(x, y) = \sum_d V_{d,t}(x, y) \tag{7}$$

The pseudo-code of ATSGM can be seen in Algorithm 1 for better understanding. Through the above steps, ATSGM is able to convert time series data into an image representation with spatial structure. This image representation can better capture the characteristics of time series data, especially the local characteristics of different channels of time series at the same time point, and provide more reliable input for subsequent image-based models.

---

**Algorithm 1** ATSGM

---

**Input**: time series $X = [X^1, X^2, ..., X^D]$ consists of $D$ different channel with $X^D = [x_1^D, x_2^D, ..., x_t^D]$, where $x_i^D$ is the value of variable $D$ at time step $i$ and the time series length is $t$

**Output**: a $224 \times 224$ matrix $N$

  1: **Resize the Time Series & Normalization**
  2: For every variable, resize its length to 196: $X^{D \times T} \to X^{D \times 196}$
  3: **Transformation**
  4: Initialize $P$ as an empty matrix with the shape of $D \times 196 \times 16 \times 16$, generate the gaussian matrix list $\Phi^{D \times 16 \times 16}$ according to the number of variable $D$
  5: **for** $i \in D$ **do**
  6:     **for** $j \in L$ **do**
  7:        $P_j^i = X_j^i \cdot \Phi_i$
  8:     **end for**
  9: **end for**
10: **Reshape** $P$
11: $P^{D \times 224 \times 224} \leftarrow P^{D \times 196 \times 16 \times 16}$
12: **Suppression** $P$ **in the dimension-0**
13: $P^{224 \times 224} \leftarrow P^{D \times 224 \times 224}$

---

[1]http://hydra.nat.uni-magdeburg.de/packing/csq/csq.html

### 4.3 CLASSIFICATION MODEL

Vision Transformer is a classical transformer-based image classification algorithm proposed in 2021(Dosovitskiy et al., 2021), which is prominent for its global feature extraction and long-dependency modeling capability because of multi-head attention. In our work, we adopt ViT-B/16 to do the image classification task with the input from our proposed time series image representation.

## 5 EXPERIMENT

### 5.1 EXPERIMENTAL SETTING

#### 5.1.1 DATASETS

The whole UCR/UEA archive (Chen et al., 2015) is utilized to test the performance of our proposed method, which includes 128 UTS Datasets and 30 MTS Datasets. This archive is a well-known and widely used classic public dataset in time series classification. It contains 158 time series datasets in total covering different scenarios with predefined train/test split, including 128 UTS Datasets and 30 MTS Datasets. Moreover, the number of classes in this archive ranges from 2 to 60. In addition, there are 4 MTS Datasets that have unequal lengths in different channels. The summary of these datasets can be seen in Appendix A, which shows detailed information including the size of the training and testing set, channel, length, class numbers, and domains of each dataset. By testing our algorithm on all datasets and comparing it with baseline models, the performance can be obtained for further analysis.

#### 5.1.2 BASELINES

Several comparison algorithms including SOTA methods are deployed to show the effectiveness of the proposed model. According to Ismail Fawaz et al. (2020), as for UTS, InceptionTime, FCN and ResNet achieve top 1 performance on 69.4% of the datasets by comparing 9 deep learning models, so these models are chosen as the baseline for the UTS classification task. When it comes to MTS, we choose five state-of-the-art multivariate time series classification models as our baselines: HIerarchical VotE Collective of Transformation-based Ensembles(HIVE-COTE)(Lines et al.), Canonical Interval Forest (CIF)(Middlehurst et al., 2020), RandOm Convolutional KErnel Transform (ROCKET)(Dempster et al., 2020), InceptionTime(Ismail Fawaz et al., 2020) and ResNet(He et al., 2016). HIVE-COTE, CIF, ROCKET, and InceptionTime, which are more accurate than other classifiers experimented on in the UEA archive by Ruiz et al. (2021). To show the effectiveness of the ATSGM of our framework, we also conducted the experiment to replace our following classifier from ViT to ResNet to find out the performance of the current two typical classification architectures from computer vision.

#### 5.1.3 IMPLEMENTATION

ViT-B/16 is adopted as the following classifier for time series image representation. Therefore, the length of all time series data equals 196 (L=196). For MTS, we set the circle area of each channel to encompass the information within a 2-standard-deviation range of the predefined 2D Gaussian distribution derived from section 3, that is to say, $\sigma$ = R/2 according to section 4. Moreover, we stick to the original training and testing set split for all datasets. All the test datasets were trained for 200 epochs. In the meantime, the value of hyper-parameters from ViT is set by default according to Dosovitskiy et al. (2021). The experiment of Time2Image is replicated for 5 times of each dataset with different random seeds and the value of the random seed is 0,1,2,3 and 4.

#### 5.1.4 EVALUATION INDICATOR

We use accuracy through 5 replicate tests and calculate the average as our evaluation indicator for performance evaluation so as to make the comparison between our proposed method and the baseline models.

## 5.2 PERFORMANCE ANALYSIS

We did extensive experiments on the whole UCR/UEA Archive and the experimental result will be analyzed in this section. Due to page limitations, the classification accuracy of all data sets will be fully disclosed in Appendix B. The corresponding critical difference diagrams are drawn based on the performance of each dataset, which illustrates multiple pieces of information that can help make a comparison of the performance of different algorithms on multiple datasets and are shown in Figure 3 and Figure 4. As for the performance comparison between Time2Image and baselines, it can be seen that our proposed framework has the best performance on both UTS and MTS datasets, indicating the generalization ability of the proposed algorithm. Moreover, Time2Image significantly outperforms other baselines with an average rank of 1.8945 in the UTS Dataset, which wins on 73 problems out of 128 and significantly outperforms ResNet from Table 1. In addition, the performance of MTS also achieved top 1 performance compared with other baselines.

Table 1: Number of different time series image representation algorithms

| Data Type | Total # | Win_# Time2Image | Win_# FCN | Win_# ResNet | Win_# ROCKET | Win_# CIF | Win_# HIVE-COTE | Win_# InceptionTime |
|---|---|---|---|---|---|---|---|---|
| UTS | 128 | **73** | 12 | 41 | | | | |
| MTS | 30 | **13** | | 3 | 4 | 3 | 2 | 5 |

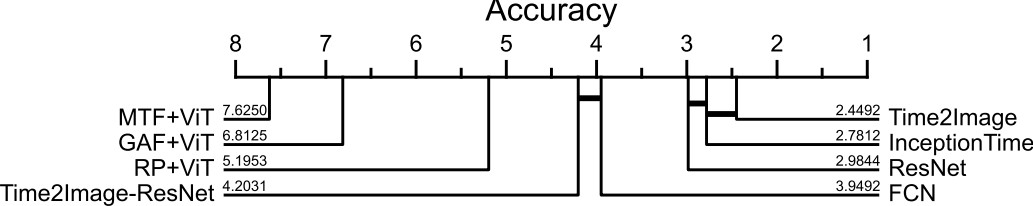

Figure 3: Critical difference diagram of UTS Dataset

Since there are some existing time series image representation methods, we also did comparison experiments on different time series image representations. GAF, MTF, and RP are universally adopted image representation methods of UTS, so we chose them for comparison, and the result can be seen in Figure 3. From the figure, it can be seen that none of the existing image representation methods can defeat baseline models. This indicates a huge research gap for time series representation for TSC, which is consistent with the current research status, but our proposed method is significantly better than not only other image representation methods but also all baselines, which provides an alternative TSC algorithm and showing a promising direction on time series image representation and providing an alternative solution on TSC task.

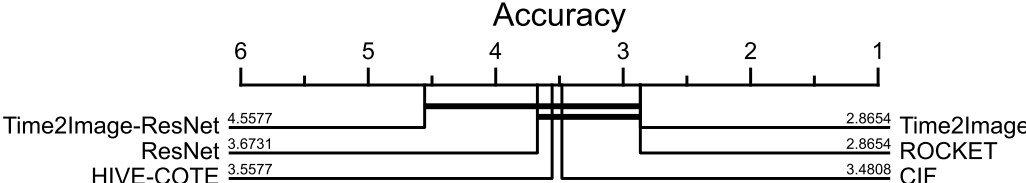

Figure 4: Critical difference diagram of MTS Dataset

In addition, to explore whether the choice of different image classification models will impact the performance, we also did an experiment on ResNet, which is a typical CNN architecture model, to replace ViT for comparison. According to the result in Figure 3, it can be seen that our proposed

framework is better than all other image representation models but not as good as SOTA, which illustrates the importance of long-range information for temporal classification and the superiority of ViT in capturing long-range information. Nevertheless, the ATSGM method we proposed still has significant advantages over other image representation learning for time series image representation, which also explains the effectiveness of our proposed ATSGM method to a certain extent.

Table 2: Classification results grouped by domains

| Category | Time2Image | FCN | ResNet | Time2Image_Win | FCN_Win | ResNet_Win |
|---|---|---|---|---|---|---|
| Device(9) | **75.96%** | 70.91% | 71.16% | **4** | 3 | 2 |
| ECG(6) | 94.67% | 92.91% | **94.98%** | 2 | 1 | **3** |
| EOG(2) | **57.98%** | 42.85% | 55.06% | **1** | 0 | **1** |
| EPG(2) | 99.76% | **100.00%** | **100.00%** | 0 | **1** | **1** |
| Hemodynamics(3) | **83.32%** | 36.63% | 62.79% | 1 | 0 | **2** |
| HRM(1) | **99.68%** | 78.06% | 98.49% | **1** | 0 | 0 |
| Image(32) | **83.32%** | 78.16% | 82.89% | **17** | 1 | 14 |
| Motion(17) | **81.99%** | 78.03% | 81.91% | **8** | 2 | 7 |
| Power(1) | **98.22%** | 90.00% | 88.89% | **1** | 0 | 0 |
| Sensor(30) | **84.26%** | 60.73% | 63.73% | **21** | 1 | 8 |
| Simulated(8) | 94.91% | 88.79% | **98.14%** | **4** | 3 | 1 |
| Spectro(8) | **84.67%** | 66.80% | 81.13% | **5** | 0 | 3 |
| Spectrum(4) | **79.89%** | 52.44% | 62.28% | **4** | 0 | 0 |
| Traffic(2) | **94.36%** | 54.06% | 54.03% | **2** | 0 | 0 |
| Trajectory(3) | **59.90%** | 55.61% | 56.33% | **2** | 0 | 1 |

To test whether it can be regarded as a unified framework, performance grouped by different domains is also conducted to find out the generalization of the model. Table 2 shows the algorithms' performance with respect to the domain of the datasets. We take the domains defined by Bagnall et al. (2017) for UTS Datasets. From the table, it can be concluded that 128 datasets can be categorized into 15 domains. The first 3 columns show the average accuracy between Time2Image and baselines within the same domain and the remaining columns calculate the winning number of datasets for each model. From the table, it can be obtained that Time2Image achieves top 1 performance on 12 out of 15 domains, indicating the inherent generalization ability of Time2Image.

## 5.3 PARAMETER ANALYSIS

From the methodology, it can be seen that our methodology is an adaptive algorithm, that is to say, the parameter, especially the value of the standard deviation($\sigma$) of Gaussian distribution seems to have an impact on the performance. In order to explore the influence of the value of the standard deviation on the performance of the model, we record the accuracy of all data sets with different standard deviation values which can be seen in Appendix C. Here we calculate the mean of the values from Appendix C of the whole datasets to indicate the final performance of the model and the results are shown in Figure 5. From the result, it can be concluded that when $\sigma = \frac{R}{2}$, the performance of the model is the best, but the difference is not that large, of which the variance is 0.37 on average, indicating the robustness of our proposed algorithms.

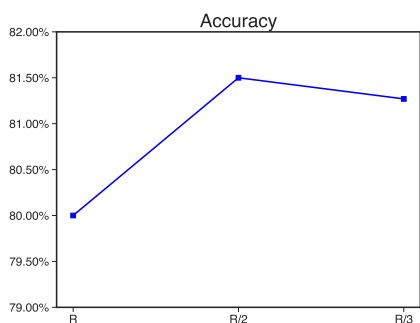

Figure 5: Parameter analysis of $\sigma$

## 6 CONCLUSION

In this work, a general time series image representation algorithm (Time2Image) was proposed, which is not only suitable for both UTS and MTS but also does a good job on non-stationary and unequal-length data. We validate the effectiveness of our approach based on all 158 public datasets from UCR/UEA. Through extensive experiments, our approach achieves notably better performance

when compared with SOTA baselines, which could be a potential solution for future time series images.

ACKNOWLEDGMENTS

We would like to express our sincere gratitude to all the reviewers and the public for your time and interest in our work. We welcome all valuable feedback and suggestions on our paper, and we think any insightful comments and constructive critiques can make this paper better.

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

# A DATASET DESCRIPTION

## A.1 SUMMARY OF UCR UNIVARIATE DATASETS

Table 3: Summary of UCR Univariate Datasets

| ID | Domain | Name | TrainSize | TestSize | Class | Length |
|---|---|---|---|---|---|---|
| 1 | Image | Adiac | 390 | 391 | 37 | 176 |
| 2 | Image | ArrowHead | 36 | 175 | 3 | 251 |
| 3 | Spectro | Beef | 30 | 30 | 5 | 470 |
| 4 | Image | BeetleFly | 20 | 20 | 2 | 512 |
| 5 | Image | BirdChicken | 20 | 20 | 2 | 512 |
| 6 | Sensor | Car | 60 | 60 | 4 | 577 |
| 7 | Simulated | CBF | 30 | 900 | 3 | 128 |
| 8 | Sensor | ChlorineConcentration | 467 | 3840 | 3 | 166 |
| 9 | Sensor | CinCECGTorso | 40 | 1380 | 4 | 1639 |
| 10 | Spectro | Coffee | 28 | 28 | 2 | 286 |
| 11 | Device | Computers | 250 | 250 | 2 | 720 |
| 12 | Motion | CricketX | 390 | 390 | 12 | 300 |
| 13 | Motion | CricketY | 390 | 390 | 12 | 300 |
| 14 | Motion | CricketZ | 390 | 390 | 12 | 300 |
| 15 | Image | DiatomSizeReduction | 16 | 306 | 4 | 345 |
| 16 | Image | DistalPhalanxOutlineAgeGroup | 400 | 139 | 3 | 80 |
| 17 | Image | DistalPhalanxOutlineCorrect | 600 | 276 | 2 | 80 |
| 18 | Image | DistalPhalanxTW | 400 | 139 | 6 | 80 |
| 19 | Sensor | Earthquakes | 322 | 139 | 2 | 512 |
| 20 | ECG | ECG200 | 100 | 100 | 2 | 96 |
| 21 | ECG | ECG5000 | 500 | 4500 | 5 | 140 |
| 22 | ECG | ECGFiveDays | 23 | 861 | 2 | 136 |
| 23 | Device | ElectricDevices | 8926 | 7711 | 7 | 96 |
| 24 | Image | FaceAll | 560 | 1690 | 14 | 131 |
| 25 | Image | FaceFour | 24 | 88 | 4 | 350 |
| 26 | Image | FacesUCR | 200 | 2050 | 14 | 131 |
| 27 | Image | FiftyWords | 450 | 455 | 50 | 270 |
| 28 | Image | Fish | 175 | 175 | 7 | 463 |
| 29 | Sensor | FordA | 3601 | 1320 | 2 | 500 |
| 30 | Sensor | FordB | 3636 | 810 | 2 | 500 |
| 31 | Motion | GunPoint | 50 | 150 | 2 | 150 |
| 32 | Spectro | Ham | 109 | 105 | 2 | 431 |
| 33 | Image | HandOutlines | 1000 | 370 | 2 | 2709 |
| 34 | Motion | Haptics | 155 | 308 | 5 | 1092 |
| 35 | Image | Herring | 64 | 64 | 2 | 512 |
| 36 | Motion | InlineSkate | 100 | 550 | 7 | 1882 |
| 37 | Sensor | InsectWingbeatSound | 220 | 1980 | 11 | 256 |
| 38 | Sensor | ItalyPowerDemand | 67 | 1029 | 2 | 24 |
| 39 | Device | LargeKitchenAppliances | 375 | 375 | 3 | 720 |
| 40 | Sensor | Lightning2 | 60 | 61 | 2 | 637 |
| 41 | Sensor | Lightning7 | 70 | 73 | 7 | 319 |
| 42 | Simulated | Mallat | 55 | 2345 | 8 | 1024 |
| 43 | Spectro | Meat | 60 | 60 | 3 | 448 |
| 44 | Image | MedicalImages | 381 | 760 | 10 | 99 |
| 45 | Image | MiddlePhalanxOutlineAgeGroup | 400 | 154 | 3 | 80 |
| 46 | Image | MiddlePhalanxOutlineCorrect | 600 | 291 | 2 | 80 |
| 47 | Image | MiddlePhalanxTW | 399 | 154 | 6 | 80 |
| 48 | Sensor | MoteStrain | 20 | 1252 | 2 | 84 |
| 49 | ECG | NonInvasiveFetalECGThorax1 | 1800 | 1965 | 42 | 750 |
| 50 | ECG | NonInvasiveFetalECGThorax2 | 1800 | 1965 | 42 | 750 |
| 51 | Spectro | OliveOil | 30 | 30 | 4 | 570 |
| 52 | Image | OSULeaf | 200 | 242 | 6 | 427 |

| 53 | Image | PhalangesOutlinesCorrect | 1800 | 858 | 2 | 80 |
|----|-------|--------------------------|------|-----|---|-----|
| 54 | Sensor | Phoneme | 214 | 1896 | 39 | 1024 |
| 55 | Sensor | Plane | 105 | 105 | 7 | 144 |
| 56 | Image | ProximalPhalanxOutlineAgeGroup | 400 | 205 | 3 | 80 |
| 57 | Image | ProximalPhalanxOutlineCorrect | 600 | 291 | 2 | 80 |
| 58 | Image | ProximalPhalanxTW | 400 | 205 | 6 | 80 |
| 59 | Device | RefrigerationDevices | 375 | 375 | 3 | 720 |
| 60 | Device | ScreenType | 375 | 375 | 3 | 720 |
| 61 | Simulated | ShapeletSim | 20 | 180 | 2 | 500 |
| 62 | Image | ShapesAll | 600 | 600 | 60 | 512 |
| 63 | Device | SmallKitchenAppliances | 375 | 375 | 3 | 720 |
| 64 | Sensor | SonyAIBORobotSurface1 | 20 | 601 | 2 | 70 |
| 65 | Sensor | SonyAIBORobotSurface2 | 27 | 953 | 2 | 65 |
| 66 | Sensor | StarLightCurves | 1000 | 8236 | 3 | 1024 |
| 67 | Spectro | Strawberry | 613 | 370 | 2 | 235 |
| 68 | Image | SwedishLeaf | 500 | 625 | 15 | 128 |
| 69 | Image | Symbols | 25 | 995 | 6 | 398 |
| 70 | Simulated | SyntheticControl | 300 | 300 | 6 | 60 |
| 71 | Motion | ToeSegmentation1 | 40 | 228 | 2 | 277 |
| 72 | Motion | ToeSegmentation2 | 36 | 130 | 2 | 343 |
| 73 | Sensor | Trace | 100 | 100 | 4 | 275 |
| 74 | ECG | TwoLeadECG | 23 | 1139 | 2 | 82 |
| 75 | Simulated | TwoPatterns | 1000 | 4000 | 4 | 128 |
| 76 | Motion | UWaveGestureLibraryAll | 896 | 3582 | 8 | 945 |
| 77 | Motion | UWaveGestureLibraryX | 896 | 3582 | 8 | 315 |
| 78 | Motion | UWaveGestureLibraryY | 896 | 3582 | 8 | 315 |
| 79 | Motion | UWaveGestureLibraryZ | 896 | 3582 | 8 | 315 |
| 80 | Sensor | Wafer | 1000 | 6164 | 2 | 152 |
| 81 | Spectro | Wine | 57 | 54 | 2 | 234 |
| 82 | Image | WordSynonyms | 267 | 638 | 25 | 270 |
| 83 | Motion | Worms | 181 | 77 | 5 | 900 |
| 84 | Motion | WormsTwoClass | 181 | 77 | 2 | 900 |
| 85 | Image | Yoga | 300 | 3000 | 2 | 426 |
| 86 | Device | ACSF1 | 100 | 100 | 10 | 1460 |
| 87 | Sensor | AllGestureWiimoteX | 300 | 700 | 10 | Vary |
| 88 | Sensor | AllGestureWiimoteY | 300 | 700 | 10 | Vary |
| 89 | Sensor | AllGestureWiimoteZ | 300 | 700 | 10 | Vary |
| 90 | Simulated | BME | 30 | 150 | 3 | 128 |
| 91 | Traffic | Chinatown | 20 | 343 | 2 | 24 |
| 92 | Image | Crop | 7200 | 16800 | 24 | 46 |
| 93 | Sensor | DodgerLoopDay | 78 | 80 | 7 | 288 |
| 94 | Sensor | DodgerLoopGame | 20 | 138 | 2 | 288 |
| 95 | Sensor | DodgerLoopWeekend | 20 | 138 | 2 | 288 |
| 96 | EOG | EOGHorizontalSignal | 362 | 362 | 12 | 1250 |
| 97 | EOG | EOGVerticalSignal | 362 | 362 | 12 | 1250 |
| 98 | Spectro | EthanolLevel | 504 | 500 | 4 | 1751 |
| 99 | Sensor | FreezerRegularTrain | 150 | 2850 | 2 | 301 |
| 100 | Sensor | FreezerSmallTrain | 28 | 2850 | 2 | 301 |
| 101 | HRM | Fungi | 18 | 186 | 18 | 201 |
| 102 | Trajectory | GestureMidAirD1 | 208 | 130 | 26 | Vary |
| 103 | Trajectory | GestureMidAirD2 | 208 | 130 | 26 | Vary |
| 104 | Trajectory | GestureMidAirD3 | 208 | 130 | 26 | Vary |
| 105 | Sensor | GesturePebbleZ1 | 132 | 172 | 6 | Vary |
| 106 | Sensor | GesturePebbleZ2 | 146 | 158 | 6 | Vary |
| 107 | Motion | GunPointAgeSpan | 135 | 316 | 2 | 150 |
| 108 | Motion | GunPointMaleVersusFemale | 135 | 316 | 2 | 150 |
| 109 | Motion | GunPointOldVersusYoung | 136 | 315 | 2 | 150 |
| 110 | Device | HouseTwenty | 40 | 119 | 2 | 2000 |

| 111 | EPG | InsectEPGRegularTrain | 62 | 249 | 3 | 601 |
| 112 | EPG | InsectEPGSmallTrain | 17 | 249 | 3 | 601 |
| 113 | Traffic | MelbournePedestrian | 1194 | 2439 | 10 | 24 |
| 114 | Image | MixedShapesRegularTrain | 500 | 2425 | 5 | 1024 |
| 115 | Image | MixedShapesSmallTrain | 100 | 2425 | 5 | 1024 |
| 116 | Sensor | PickupGestureWiimoteZ | 50 | 50 | 10 | Vary |
| 117 | Hemodynamics | PigAirwayPressure | 104 | 208 | 52 | 2000 |
| 118 | Hemodynamics | PigArtPressure | 104 | 208 | 52 | 2000 |
| 119 | Hemodynamics | PigCVP | 104 | 208 | 52 | 2000 |
| 120 | Device | PLAID | 537 | 537 | 11 | Vary |
| 121 | Power | PowerCons | 180 | 180 | 2 | 144 |
| 122 | Spectrum | Rock | 20 | 50 | 4 | 2844 |
| 123 | Spectrum | SemgHandGenderCh2 | 300 | 600 | 2 | 1500 |
| 124 | Spectrum | SemgHandMovementCh2 | 450 | 450 | 6 | 1500 |
| 125 | Spectrum | SemgHandSubjectCh2 | 450 | 450 | 5 | 1500 |
| 126 | Sensor | ShakeGestureWiimoteZ | 50 | 50 | 10 | Vary |
| 127 | Simulated | SmoothSubspace | 150 | 150 | 3 | 15 |
| 128 | Simulated | UMD | 36 | 144 | 3 | 150 |

## A.2 SUMMARY OF UEA MULTIVARIATE DATASETS

Table 4: Summary of UEA Multivariate Datasets

| ID | Dataset | TrainSize | TestSize | NumDimensions | Class | Length |
|----|---------|-----------|----------|---------------|-------|--------|
| 1 | ArticularyWordRecognition | 275 | 300 | 9 | 25 | 144 |
| 2 | AtrialFibrillation | 15 | 15 | 2 | 3 | 640 |
| 3 | BasicMotions | 40 | 40 | 6 | 4 | 100 |
| 4 | CharacterTrajectories | 1422 | 1436 | 3 | 20 | 182 |
| 5 | Cricket | 108 | 72 | 6 | 12 | 1197 |
| 6 | DuckDuckGeese | 50 | 50 | 1345 | 5 | 270 |
| 7 | EigenWorms | 128 | 131 | 6 | 5 | 17984 |
| 8 | Epilepsy | 137 | 138 | 3 | 4 | 206 |
| 9 | EthanolConcentration | 261 | 263 | 3 | 4 | 1751 |
| 10 | ERing | 30 | 270 | 4 | 6 | 65 |
| 11 | FaceDetection | 5890 | 3524 | 144 | 2 | 62 |
| 12 | FingerMovements | 316 | 100 | 28 | 2 | 50 |
| 13 | HandMovementDirection | 160 | 74 | 10 | 4 | 400 |
| 14 | Handwriting | 150 | 850 | 3 | 26 | 152 |
| 15 | Heartbeat | 204 | 205 | 61 | 2 | 405 |
| 16 | InsectWingbeat | 30000 | 20000 | 200 | 10 | 30 |
| 17 | JapaneseVowels | 270 | 370 | 12 | 9 | 29 |
| 18 | Libras | 180 | 180 | 2 | 15 | 45 |
| 19 | LSST | 2459 | 2466 | 6 | 14 | 36 |
| 20 | MotorImagery | 278 | 100 | 64 | 2 | 3000 |
| 21 | NATOPS | 180 | 180 | 24 | 6 | 51 |
| 22 | PenDigits | 7494 | 3498 | 2 | 10 | 8 |
| 23 | PEMS-SF | 267 | 173 | 963 | 7 | 144 |
| 24 | Phoneme | 3315 | 3353 | 11 | 39 | 217 |
| 25 | RacketSports | 151 | 152 | 6 | 4 | 30 |
| 26 | SelfRegulationSCP1 | 268 | 293 | 6 | 2 | 896 |
| 27 | SelfRegulationSCP2 | 200 | 180 | 7 | 2 | 1152 |
| 28 | SpokenArabicDigits | 6599 | 2199 | 13 | 10 | 93 |
| 29 | StandWalkJump | 12 | 15 | 4 | 3 | 2500 |
| 30 | UWaveGestureLibrary | 120 | 320 | 3 | 8 | 315 |

## B PERFORMANCE OF UCR/UEA DATASETS IN DETAIL

Table 5: Classification Results of UCR Univariate Time Series Datasets

| Dataset | Time2Image | InceptionTime | FCN | ResNet | GAF+ViT | MTF+ViT | RP+ViT | Time2Image-ResNet |
|---|---|---|---|---|---|---|---|---|
| ACSF1 | 82.60% | 91.00% | 85.20% | 88.80% | 14.00% | 10.00% | 36.00% | 44.80% |
| Adiac | 77.95% | 84.40% | 71.30% | 84.81% | 9.72% | 3.07% | 28.90% | 72.58% |
| AllGestureWiimoteX | 70.97% | - | 10.00% | 10.00% | - | - | - | 59.57% |
| AllGestureWiimoteY | 72.06% | - | 10.00% | 10.00% | - | - | - | 65.20% |
| AllGestureWiimoteZ | 65.29% | - | 10.00% | 10.00% | - | - | - | 54.69% |
| ArrowHead | 84.46% | 86.29% | 66.06% | 83.09% | 40.57% | 39.43% | 64.00% | 54.74% |
| Beef | 91.33% | 66.67% | 47.33% | 64.00% | 40.00% | 33.33% | 43.33% | 52.80% |
| BeetleFly | 99.00% | 85.00% | 78.00% | 83.00% | 75.00% | 50.00% | 95.00% | 63.00% |
| BirdChicken | 100.00% | 95.00% | 91.00% | 93.00% | 65.00% | 50.00% | 80.00% | 74.00% |
| BME | 100.00% | 100.00% | 56.40% | 96.80% | 50.67% | 33.33% | 52.67% | 93.27% |
| Car | 83.33% | 90.00% | 70.67% | 92.33% | 28.33% | 23.33% | 51.67% | 60.36% |
| CBF | 99.96% | 99.89% | 98.71% | 99.62% | 41.33% | 33.11% | 80.67% | 77.27% |
| Chinatown | 98.13% | 98.54% | 98.08% | 98.02% | 76.09% | 72.59% | 92.71% | 79.77% |
| ChlorineConcentration | 80.04% | 87.60% | 65.11% | 84.28% | 54.71% | 53.26% | 55.99% | 56.92% |
| CinCECGTorso | 77.88% | 85.29% | 70.29% | 75.94% | 42.46% | 24.86% | 95.07% | 37.65% |
| Coffee | 100.00% | 100.00% | 99.29% | 100.00% | 78.57% | 53.57% | 92.86% | 77.14% |
| Computers | 79.28% | 80.80% | 84.40% | 83.68% | 67.60% | 50.80% | 65.60% | 68.64% |
| CricketX | 79.95% | 84.87% | 74.46% | 78.87% | 14.62% | 7.44% | 62.56% | 74.41% |
| CricketY | 79.08% | 84.36% | 75.13% | 80.56% | 16.41% | 7.95% | 63.85% | 75.23% |
| CricketZ | 84.10% | 84.87% | 77.33% | 80.87% | 14.87% | 7.18% | 60.77% | 75.64% |
| Crop | 75.00% | 79.85% | 75.68% | 76.81% | 29.40% | 4.17% | 4.17% | 75.24% |
| DiatomSizeReduction | 98.37% | 94.77% | 79.28% | 93.99% | 46.08% | 30.07% | 54.25% | 67.19% |
| DistalPhalanxOutlineAgeGroup | 79.14% | 73.38% | 76.69% | 72.95% | 58.99% | 46.76% | 78.42% | 77.41% |
| DistalPhalanxOutlineCorrect | 81.59% | 78.62% | 76.88% | 77.83% | 63.77% | 58.33% | 75.36% | 79.20% |
| DistalPhalanxTW | 72.52% | 65.47% | 70.22% | 68.49% | 46.76% | 30.22% | 69.06% | 73.67% |
| DodgerLoopDay | 67.00% | - | 15.75% | 15.75% | - | - | - | 65.75% |
| DodgerLoopGame | 94.49% | - | 51.30% | 48.70% | - | - | - | 61.30% |
| DodgerLoopWeekend | 98.55% | - | 54.78% | 64.35% | - | - | - | 87.39% |
| Earthquakes | 78.71% | 71.22% | 72.52% | 73.53% | 76.98% | 74.82% | 74.82% | 78.14% |
| ECG200 | 93.40% | 91.00% | 88.80% | 86.80% | 73.00% | 64.00% | 86.00% | 87.68% |
| ECG5000 | 94.82% | 94.16% | 94.02% | 93.91% | 78.58% | 58.38% | 94.44% | 88.18% |
| ECGFiveDays | 91.87% | 100.00% | 93.64% | 97.96% | 67.94% | 49.71% | 79.67% | 63.22% |
| ElectricDevices | 76.86% | 72.12% | 74.69% | 72.03% | | | | 69.93% |

Table 5: Classification Results of UCR Univariate Time Series Datasets

| Dataset | Time2Image | InceptionTime | FCN | ResNet | GAF+ViT | MTF+ViT | RP+ViT | Time2Image-ResNet |
|---|---|---|---|---|---|---|---|---|
| EOGHorizontalSignal | 60.61% | 62.98% | 48.40% | 64.36% | 11.05% | 8.56% | 46.13% | 58.17% |
| EOGVerticalSignal | 55.36% | 51.66% | 37.29% | 45.75% | 9.39% | 8.84% | 41.44% | 60.06% |
| EthanolLevel | 65.16% | 83.20% | 42.20% | 85.52% | 25.80% | 25.20% | 29.80% | 29.88% |
| FaceAll | 74.72% | 79.29% | 91.01% | 85.65% | 28.82% | 16.98% | 79.41% | 79.20% |
| FaceFour | 88.41% | 95.45% | 75.91% | 94.77% | 27.27% | 29.55% | 61.36% | 38.86% |
| FacesUCR | 90.64% | 97.22% | 91.98% | 94.73% | 27.46% | 14.34% | 72.34% | 87.44% |
| FiftyWords | 79.12% | 84.62% | 53.49% | 71.91% | 23.52% | 12.53% | 66.59% | 78.20% |
| Fish | 90.17% | 98.29% | 90.63% | 98.17% | 29.71% | 12.57% | 53.14% | 94.74% |
| FordA | 93.98% | 96.06% | 91.89% | 93.20% | - | - | - | 94.97% |
| FordB | 80.86% | 84.94% | 79.46% | 81.14% | - | - | - | 81.60% |
| FreezerRegularTrain | 99.59% | 99.68% | 99.47% | 99.82% | 70.42% | 50.00% | 78.84% | 91.24% |
| FreezerSmallTrain | 90.89% | 84.60% | 74.89% | 83.17% | 62.28% | 50.00% | 78.14% | 77.35% |
| Fungi | 99.68% | - | 78.06% | 98.49% | 13.98% | 10.22% | 74.19% | 36.24% |
| GestureMidAirD1 | 72.77% | - | 69.23% | 68.00% | - | - | - | 71.69% |
| GestureMidAirD2 | 66.46% | - | 64.61% | 67.00% | - | - | - | 68.46% |
| GestureMidAirD3 | 40.46% | - | 33.00% | 34.00% | - | - | - | 44.46% |
| GesturePebbleZ1 | 95.12% | - | 15.70% | 15.70% | - | - | - | 90.93% |
| GesturePebbleZ2 | 85.82% | - | 15.82% | 15.82% | - | - | - | 87.47% |
| GunPoint | 99.60% | 100.00% | 99.33% | 100.00% | 72.00% | 50.67% | 77.33% | 65.60% |
| GunPointAgeSpan | 98.73% | 99.05% | 90.57% | 99.37% | 73.42% | 50.63% | 99.05% | 96.90% |
| GunPointMaleVersusFemale | 99.81% | 100.00% | 99.81% | 99.49% | 69.94% | 52.53% | 99.68% | 99.18% |
| GunPointOldVersusYoung | 99.49% | 100.00% | 100.00% | 100.00% | 59.37% | 52.38% | 100.00% | 89.24% |
| Ham | 79.43% | 72.38% | 73.71% | 75.43% | 57.14% | 51.43% | 62.86% | 69.71% |
| HandOutlines | 92.86% | 95.95% | 77.35% | 92.97% | 66.22% | 64.05% | 92.16% | 92.38% |
| Haptics | 50.97% | 57.14% | 42.79% | 53.25% | 20.78% | 20.78% | 43.83% | 30.91% |
| Herring | 70.00% | 70.31% | 64.37% | 65.63% | 59.38% | 59.38% | 59.38% | 60.63% |
| HouseTwenty | 95.46% | 91.60% | 91.93% | 95.29% | 65.55% | 57.98% | 88.24% | 79.16% |
| InlineSkate | 40.58% | 48.55% | 35.89% | 46.40% | 16.36% | 16.36% | 26.18% | 27.13% |
| InsectEPGRegularTrain | 100.00% | 100.00% | 100.00% | 100.00% | 81.12% | 47.39% | 47.39% | 86.43% |
| InsectEPGSmallTrain | 99.52% | 100.00% | 100.00% | 100.00% | 67.87% | 47.39% | 47.39% | 83.21% |
| InsectWingbeatSound | 59.80% | 63.69% | 40.90% | 50.31% | 19.65% | 9.19% | 58.23% | 62.21% |
| ItalyPowerDemand | 96.85% | 96.60% | 95.86% | 95.72% | 68.80% | 50.15% | 96.02% | 85.15% |
| LargeKitchenAppliances | 85.55% | 90.40% | 89.76% | 90.08% | 48.53% | 34.40% | 74.67% | 75.31% |

Table 5: Classification Results of UCR Univariate Time Series Datasets

| Dataset | Time2Image | InceptionTime | FCN | ResNet | GAF+ViT | MTF+ViT | RP+ViT | Time2Image-ResNet |
|---|---|---|---|---|---|---|---|---|
| Lightning2 | 85.25% | 83.61% | 71.48% | 75.74% | - | - | - | 79.67% |
| Lightning7 | 83.84% | 79.45% | 71.78% | 80.00% | 36.99% | 26.03% | 65.75% | 75.89% |
| Mallat | 93.93% | 95.78% | 72.95% | 96.34% | 12.41% | 12.41% | 52.79% | 73.09% |
| Meat | 92.67% | 95.00% | 75.00% | 93.33% | 41.67% | 33.33% | 71.67% | 42.33% |
| MedicalImages | 78.16% | 79.61% | 77.45% | 77.74% | 52.50% | 51.45% | 72.24% | 70.47% |
| MelbournePedestrian | 90.59% | - | 10.05% | 10.05% | - | - | - | 91.63% |
| MiddlePhalanxOutlineAgeGroup | 66.75% | 55.19% | 59.74% | 54.42% | 43.51% | 18.83% | 63.64% | 65.71% |
| MiddlePhalanxOutlineCorrect | 82.41% | 83.51% | 82.34% | 81.51% | 63.57% | 57.04% | 68.38% | 67.56% |
| MiddlePhalanxTW | 64.29% | 53.25% | 55.19% | 48.57% | 46.75% | 27.27% | 61.04% | 61.82% |
| MixedShapesRegularTrain | 92.92% | 97.03% | 94.93% | 97.68% | 26.97% | 26.97% | 90.89% | 90.92% |
| MixedShapesSmallTrain | 85.76% | 91.09% | 87.92% | 92.29% | 26.97% | 26.97% | 79.18% | 83.99% |
| MoteStrain | 91.82% | 89.38% | 91.63% | 92.16% | 59.82% | 54.95% | 82.35% | 71.90% |
| NonInvasiveFetalECGThorax1 | 94.14% | 95.83% | 89.97% | 96.04% | 11.40% | 2.34% | 84.89% | 92.73% |
| NonInvasiveFetalECGThorax2 | 94.05% | 95.88% | 91.09% | 95.29% | 26.46% | 2.34% | 89.67% | 93.82% |
| OliveOil | 74.67% | 86.67% | 40.00% | 62.00% | 40.00% | 40.00% | 40.00% | 44.66% |
| OSULeaf | 76.20% | 92.15% | 97.27% | 98.43% | 26.03% | 18.18% | 77.27% | 58.08% |
| PhalangesOutlinesCorrect | 83.85% | 85.20% | 80.82% | 82.73% | 62.94% | 61.31% | 67.72% | 80.11% |
| Phoneme | 26.34% | 33.70% | 32.75% | 35.31% | 11.29% | 11.29% | 19.78% | 30.58% |
| PickupGestureWiimoteZ | 72.40% | - | 10.00% | 10.00% | - | - | - | 48.16% |
| PigAirwayPressure | 23.94% | 94.23% | 31.15% | 67.50% | 3.37% | 1.92% | 8.65% | 24.96% |
| PigArtPressure | 72.12% | 100.00% | 58.27% | 99.52% | 2.40% | 1.92% | 61.06% | 43.75% |
| PigCVP | 69.04% | 96.63% | 20.48% | 21.35% | 1.92% | 1.92% | 16.35% | 41.15% |
| PLAID | 81.53% | - | 16.20% | 16.20% | - | - | - | 71.98% |
| Plane | 100.00% | 100.00% | 100.00% | 100.00% | 43.81% | 9.52% | 100.00% | 99.43% |
| PowerCons | 98.22% | 99.44% | 90.00% | 88.89% | 86.67% | 50.00% | 95.00% | 97.00% |
| ProximalPhalanxOutlineAgeGroup | 88.10% | 84.88% | 84.68% | 84.19% | 69.27% | 85.37% | 87.80% | 88.59% |
| ProximalPhalanxOutlineCorrect | 83.37% | 93.13% | 89.35% | 90.93% | 69.42% | 68.38% | 83.85% | 87.35% |
| ProximalPhalanxTW | 83.02% | 80.00% | 81.07% | 77.56% | 56.59% | 35.12% | 81.46% | 81.95% |
| RefrigerationDevices | 56.11% | 51.47% | 52.37% | 52.32% | 49.33% | 33.87% | 56.00% | 57.17% |
| Rock | 79.20% | 60.00% | 31.60% | 55.20% | 42.00% | 42.00% | 80.00% | 49.20% |
| ScreenType | 52.21% | 59.47% | 63.57% | 63.31% | 45.07% | 33.60% | 42.67% | 46.29% |
| SemgHandGenderCh2 | 90.80% | 87.33% | 80.77% | 83.77% | 65.00% | 65.00% | 94.17% | 89.87% |
| SemgHandMovementCh2 | 64.67% | 58.44% | 44.00% | 50.62% | 16.67% | 16.67% | 72.67% | 50.36% |

Table 5: Classification Results of UCR Univariate Time Series Datasets

| Dataset | Time2Image | InceptionTime | FCN | ResNet | GAF+ViT | MTF+ViT | RP+ViT | Time2Image-ResNet |
|---|---|---|---|---|---|---|---|---|
| SemgHandSubjectCh2 | 84.89% | 73.33% | 53.38% | 59.51% | 20.00% | 20.00% | 92.89% | 81.16% |
| ShakeGestureWiimoteZ | 93.20% | - | 10.00% | 10.00% | - | - | - | 45.60% |
| ShapeletSim | 67.11% | 99.44% | 99.22% | 95.11% | 96.67% | 50.00% | 62.78% | 54.22% |
| ShapesAll | 85.33% | 92.50% | 80.90% | 90.63% | 12.83% | 1.67% | 71.17% | 83.07% |
| SmallKitchenAppliances | 74.03% | 77.87% | 80.11% | 78.72% | 53.33% | 34.13% | 70.40% | 75.41% |
| SmoothSubspace | 99.33% | 98.00% | 99.47% | 98.67% | 55.33% | 37.33% | 33.33% | 98.67% |
| SonyAIBORobotSurface1 | 92.18% | 87.85% | 95.61% | 93.41% | 49.25% | 42.93% | 95.51% | 64.66% |
| SonyAIBORobotSurface2 | 94.19% | 95.17% | 97.33% | 98.11% | 65.90% | 61.70% | 80.06% | 74.14% |
| StarLightCurves | 97.56% | 97.84% | 97.22% | 97.44% | 72.30% | 57.72% | 96.95% | 97.18% |
| Strawberry | 97.41% | 98.38% | 96.16% | 97.62% | 71.08% | 64.32% | 94.86% | 97.30% |
| SwedishLeaf | 92.74% | 96.96% | 95.97% | 96.06% | 22.40% | 5.28% | 90.24% | 91.90% |
| Symbols | 93.05% | 98.49% | 82.93% | 96.34% | 39.20% | 17.39% | 52.66% | 78.83% |
| SyntheticControl | 99.93% | 99.67% | 98.33% | 99.80% | 31.00% | 16.67% | 66.67% | 99.33% |
| ToeSegmentation1 | 95.26% | 96.49% | 95.79% | 96.14% | 54.82% | 52.63% | 74.56% | 80.70% |
| ToeSegmentation2 | 92.46% | 94.62% | 91.38% | 89.23% | 82.31% | 81.54% | 90.77% | 87.38% |
| Trace | 100.00% | 100.00% | 100.00% | 100.00% | 56.00% | 19.00% | 78.00% | 96.60% |
| TwoLeadECG | 99.72% | 99.56% | 99.93% | 99.89% | 58.21% | 50.04% | 67.52% | 63.51% |
| TwoPatterns | 100.00% | 100.00% | 85.97% | 99.90% | 41.43% | 25.88% | 51.13% | 100.00% |
| UMD | 99.03% | 98.61% | 99.31% | 98.89% | 48.61% | 33.33% | 52.78% | 89.86% |
| UWaveGestureLibraryAll | 96.90% | 95.09% | 79.97% | 86.74% | 25.43% | 12.53% | 90.87% | 96.71% |
| UWaveGestureLibraryX | 82.57% | 82.22% | 77.84% | 78.71% | 25.54% | 12.53% | 64.32% | 81.38% |
| UWaveGestureLibraryY | 74.33% | 77.14% | 66.59% | 68.34% | 27.44% | 18.45% | 66.16% | 72.16% |
| UWaveGestureLibraryZ | 76.33% | 76.86% | 74.38% | 76.24% | 27.83% | 12.20% | 67.09% | 74.47% |
| Wafer | 99.68% | 99.87% | 99.67% | 99.89% | 92.94% | 89.21% | 99.66% | 99.64% |
| Wine | 76.67% | 62.96% | 60.74% | 71.11% | 59.26% | 50.00% | 55.56% | 51.11% |
| WordSynonyms | 69.72% | 74.45% | 45.58% | 58.40% | 29.15% | 21.94% | 54.08% | 68.37% |
| Worms | 64.68% | 80.52% | 69.09% | 81.04% | 61.04% | 42.86% | 63.64% | 62.86% |
| WormsTwoClass | 78.96% | 76.62% | 76.10% | 77.14% | 64.94% | 57.14% | 79.22% | 65.71% |
| Yoga | 86.80% | 90.50% | 75.17% | 87.17% | 55.37% | 53.57% | 76.93% | 77.60% |
| Average accuracy | 87.76% | 88.84% | 81.12% | 84.25% | 49.17% | 36.90% | 73.44% | 79.07% |

Table 6: Classification Results of UEA Multivariate Time Series Datasets

| Dataset | Time2Image | ROCKET | CIF | HIVE-COTE | ResNet | InceptionTime | Time2Image+ResNet |
|---|---|---|---|---|---|---|---|
| ArticularyWordRecognition | 97.80% | 99.56% | 97.89% | 97.99% | 98.26% | 99.10% | 94.33% |
| AtrialFibrillation | 45.33% | 24.89% | 25.11% | 25.11% | 36.22% | 22.00% | 53.33% |
| BasicMotions | 99.00% | 99.00% | 99.75% | 100.00% | 100.00% | 100.00% | 58.50% |
| CharacterTrajectories | 99.50% | - | - | - | - | - | 99.25% |
| Cricket | 100.00% | 100.00% | 98.38% | 99.26% | 99.40% | 99.44% | 90.83% |
| DuckDuckGeese | 47.20% | 46.13% | 56.00% | 47.60% | 63.20% | 63.47% | 47.60% |
| ERing | 89.93% | 86.28% | 90.33% | 78.17% | 45.45% | - | 44.12% |
| EigenWorms | 79.24% | 99.08% | 98.38% | 100.00% | 99.18% | 98.65% | 74.06% |
| Epilepsy | 96.09% | 44.68% | 72.89% | 80.68% | 28.66% | 27.92% | 26.31% |
| EthanolConcentration | 28.67% | 98.05% | 95.65% | 94.26% | 87.19% | 92.10% | 55.48% |
| FaceDetection | 67.79% | 69.42% | 68.89% | 69.17% | 62.97% | 77.24% | 50.04% |
| FingerMovements | 58.00% | 55.27% | 53.90% | 53.77% | 54.70% | 56.13% | 55.00% |
| HandMovementDirection | 61.89% | 44.59% | 52.21% | 37.79% | 35.32% | 42.39% | 41.08% |
| Handwriting | 44.80% | 56.67% | 35.13% | 50.41% | 59.78% | 65.74% | 16.05% |
| Heartbeat | 75.61% | 71.76% | 76.52% | 72.18% | 63.89% | 73.20% | 72.59% |
| InsectWingbeat | 47.60% | - | - | - | - | - | 52.97% |
| JapaneseVowels | 89.51% | - | - | - | - | - | 83.30% |
| LSST | 65.24% | 90.61% | 91.67% | 90.28% | 94.11% | 88.72% | 79.89% |
| Libras | 86.44% | 63.15% | 56.17% | 53.84% | 42.94% | 33.97% | 61.01% |
| MotorImagery | 63.60% | 53.13% | 51.80% | 52.17% | 49.77% | 51.17% | 55.40% |
| NATOPS | 86.33% | 88.54% | 84.41% | 82.85% | 97.11% | 96.63% | 78.11% |
| PEMS-SF | 84.62% | 99.56% | 98.97% | 97.19% | 99.64% | 99.68% | 99.05% |
| PenDigits | 99.11% | 85.63% | 99.85% | 97.98% | 81.95% | 82.83% | 77.11% |
| PhonemeSpectra | 24.94% | 28.35% | 26.56% | 32.87% | 30.86% | 36.74% | 23.08% |
| RacketSports | 89.87% | 92.79% | 89.30% | 90.64% | 94.23% | 91.69% | 73.03% |
| SelfRegulationSCP1 | 85.94% | 86.55% | 85.94% | 86.02% | 76.11% | 84.69% | 86.14% |
| SelfRegulationSCP2 | 60.22% | 51.35% | 48.87% | 51.67% | 50.24% | 52.04% | 52.11% |
| SpokenArabicDigits | 99.41% | - | - | - | - | - | 99.32% |
| StandWalkJump | 66.67% | 45.56% | 45.11% | 40.67% | 30.89% | 42.00% | 66.67% |
| UWaveGestureLibrary | 89.31% | 94.43% | 92.42% | 91.31% | 88.35% | 91.23% | 80.31% |
| Average accuracy | 74.32% | 72.12% | 72.77% | 72.07% | 68.09% | 70.75% | 64.87% |

## C PARAMETER ANALYSIS

Table 7: Parameter Analysis of UCR/UEA Archive

| Dataset | $\sigma$=R | $\sigma$=R/2 | $\sigma$=R/3 | Difference | Variance |
|---|---|---|---|---|---|
| ArticularyWordRecognition | 97.0000% | 97.8000% | 97.6667% | 0.8000% | 0.0018% |
| AtrialFibrillation | 40.0000% | 45.3333% | 40.0000% | 5.3333% | 0.0948% |
| BasicMotions | 97.5000% | 99.0000% | 97.5000% | 1.5000% | 0.0075% |
| CharacterTrajectories | 99.3036% | 99.4986% | 99.5125% | 0.2089% | 0.0001% |
| Cricket | 98.6111% | 100.0000% | 98.6111% | 1.3889% | 0.0064% |
| DuckDuckGeese | 52.0000% | 47.2000% | 46.0000% | 6.0000% | 0.1008% |
| EigenWorms | 77.8626% | 89.9259% | 78.6260% | 12.0633% | 0.4563% |
| Epilepsy | 94.9275% | 79.2366% | 97.1015% | 17.8648% | 0.9501% |
| EthanolConcentration | 28.8973% | 96.0870% | 31.5589% | 67.1896% | 14.4757% |
| ERing | 87.0370% | 28.6692% | 94.0741% | 65.4049% | 12.8902% |
| FaceDetection | 67.2247% | 67.7923% | 67.7639% | 0.5675% | 0.0010% |
| FingerMovements | 61.0000% | 58.0000% | 59.0000% | 3.0000% | 0.0233% |
| HandMovementDirection | 50.0000% | 61.8919% | 60.8108% | 11.8919% | 0.4324% |
| Handwriting | 42.3529% | 44.8000% | 45.4118% | 3.0588% | 0.0262% |
| Heartbeat | 75.1220% | 75.6098% | 76.0976% | 0.9756% | 0.0024% |
| InsectWingbeat | 47.1480% | 47.6024% | 47.5600% | 0.4544% | 0.0006% |
| JapaneseVowels | 91.6216% | 89.5135% | 91.8919% | 2.3784% | 0.0170% |
| Libras | 86.1111% | 65.2393% | 88.3333% | 23.0941% | 1.6232% |
| LSST | 64.8824% | 86.4444% | 65.5312% | 21.5620% | 1.5045% |
| MotorImagery | 60.0000% | 63.6000% | 62.0000% | 3.6000% | 0.0325% |
| NATOPS | 85.0000% | 86.3333% | 87.7778% | 2.7778% | 0.0193% |
| PenDigits | 98.7707% | 84.6243% | 99.2281% | 14.6039% | 0.6893% |
| PEMS-SF | 86.7052% | 99.1138% | 84.3931% | 14.7207% | 0.6267% |
| PhonemeSpectra | 26.0960% | 24.9389% | 25.4399% | 1.1572% | 0.0034% |
| RacketSports | 90.1316% | 89.8684% | 88.1579% | 1.9737% | 0.0115% |
| SelfRegulationSCP1 | 87.0307% | 85.9386% | 86.0068% | 1.0921% | 0.0037% |
| SelfRegulationSCP2 | 60.0000% | 60.2222% | 55.0000% | 5.2222% | 0.0872% |
| SpokenArabicDigits | 99.4543% | 99.4088% | 99.3633% | 0.0910% | 0.0000% |
| StandWalkJump | 73.3333% | 66.6667% | 80.0000% | 13.3333% | 0.4444% |
| UWaveGestureLibrary | 90.0000% | 89.3125% | 90.3125% | 1.0000% | 0.0026% |
| ACSF1 | 74.0000% | 82.6000% | 80.0000% | 8.6000% | 0.1945% |
| Adiac | 69.8210% | 77.9540% | 77.2379% | 8.1330% | 0.2028% |
| AllGestureWiimoteX | 72.7143% | 70.9714% | 72.5714% | 1.7429% | 0.0094% |
| AllGestureWiimoteY | 71.7143% | 72.0571% | 72.0000% | 0.3429% | 0.0003% |
| AllGestureWiimoteZ | 63.2857% | 65.2857% | 65.2857% | 2.0000% | 0.0133% |
| ArrowHead | 82.2857% | 84.4571% | 82.8571% | 2.1714% | 0.0127% |
| Beef | 90.0000% | 91.3333% | 90.0000% | 1.3333% | 0.0059% |
| BeetleFly | 95.0000% | 99.0000% | 100.0000% | 5.0000% | 0.0700% |
| BirdChicken | 100.0000% | 100.0000% | 100.0000% | 0.0000% | 0.0000% |
| BME | 100.0000% | 100.0000% | 100.0000% | 0.0000% | 0.0000% |
| Car | 75.0000% | 83.3333% | 85.0000% | 10.0000% | 0.2870% |
| CBF | 99.6667% | 99.9556% | 100.0000% | 0.3333% | 0.0003% |
| Chinatown | 98.8338% | 98.1341% | 97.9592% | 0.8746% | 0.0021% |
| ChlorineConcentration | 68.9323% | 80.0365% | 81.1719% | 12.2396% | 0.4573% |
| CinCECGTorso | 78.7681% | 77.8841% | 70.9420% | 7.8261% | 0.1837% |
| Coffee | 100.0000% | 100.0000% | 100.0000% | 0.0000% | 0.0000% |
| Computers | 76.4000% | 79.2800% | 77.2000% | 2.8800% | 0.0221% |
| CricketX | 80.5128% | 79.9487% | 82.0513% | 2.1026% | 0.0118% |
| CricketY | 79.7436% | 79.0769% | 77.6923% | 2.0513% | 0.0109% |
| CricketZ | 84.6154% | 84.1026% | 83.8462% | 0.7692% | 0.0015% |
| Crop | 74.8869% | 74.9952% | 74.2381% | 0.7571% | 0.0017% |
| DiatomSizeReduction | 88.5621% | 98.3660% | 98.3660% | 9.8039% | 0.3204% |
| DistalPhalanxOutlineAgeGroup | 76.2590% | 79.1367% | 78.4173% | 2.8777% | 0.0224% |

**Table 7 continued from previous page**

| Dataset | $\sigma$=R | $\sigma$=R/2 | $\sigma$=R/3 | Difference | Variance |
|---|---|---|---|---|---|
| DistalPhalanxOutlineCorrect | 80.7971% | 81.5942% | 81.5217% | 0.7971% | 0.0019% |
| DistalPhalanxTW | 72.6619% | 72.5180% | 71.9424% | 0.7194% | 0.0014% |
| DodgerLoopDay | 63.7500% | 67.0000% | 65.0000% | 3.2500% | 0.0269% |
| DodgerLoopGame | 94.2029% | 94.4928% | 94.2029% | 0.2899% | 0.0003% |
| DodgerLoopWeekend | 97.8261% | 98.5507% | 97.8261% | 0.7246% | 0.0018% |
| Earthquakes | 76.9784% | 78.7050% | 77.6978% | 1.7266% | 0.0075% |
| ECG200 | 93.0000% | 93.4000% | 92.0000% | 1.4000% | 0.0052% |
| ECG5000 | 94.3778% | 94.8222% | 94.8222% | 0.4444% | 0.0007% |
| ECGFiveDays | 92.9152% | 91.8699% | 95.5865% | 3.7166% | 0.0367% |
| ElectricDevices | 76.7864% | 76.8590% | 76.1769% | 0.6821% | 0.0014% |
| EOGHorizontalSignal | 58.2873% | 60.6077% | 61.6022% | 3.3149% | 0.0289% |
| EOGVerticalSignal | 55.5249% | 55.3591% | 55.5249% | 0.1657% | 0.0001% |
| EthanolLevel | 30.8000% | 65.1600% | 76.0000% | 45.2000% | 5.5686% |
| FaceAll | 76.3905% | 74.7219% | 74.6746% | 1.7160% | 0.0096% |
| FaceFour | 85.2273% | 88.4091% | 89.7727% | 4.5455% | 0.0544% |
| FacesUCR | 90.4878% | 90.6439% | 89.4146% | 1.2293% | 0.0045% |
| FiftyWords | 78.9011% | 79.1209% | 80.0000% | 1.0989% | 0.0034% |
| Fish | 86.2857% | 90.1714% | 92.5714% | 6.2857% | 0.1006% |
| FordA | 94.0909% | 93.9848% | 93.8636% | 0.2273% | 0.0001% |
| FordB | 79.7531% | 80.8642% | 81.4815% | 1.7284% | 0.0077% |
| FreezerRegularTrain | 99.6842% | 99.5860% | 99.4386% | 0.2456% | 0.0002% |
| FreezerSmallTrain | 88.3860% | 90.8912% | 87.2281% | 3.6632% | 0.0351% |
| Fungi | 80.1075% | 99.6774% | 100.0000% | 19.8925% | 1.2980% |
| GestureMidAirD1 | 76.1538% | 72.7692% | 72.3077% | 3.8462% | 0.0441% |
| GestureMidAirD2 | 66.9231% | 66.4615% | 67.6923% | 1.2308% | 0.0039% |
| GestureMidAirD3 | 40.7692% | 40.4615% | 41.5385% | 1.0769% | 0.0031% |
| GesturePebbleZ1 | 95.9302% | 95.1163% | 95.3488% | 0.8140% | 0.0018% |
| GesturePebbleZ2 | 86.7089% | 85.8228% | 87.3418% | 1.5190% | 0.0058% |
| GunPoint | 100.0000% | 99.6000% | 97.3333% | 2.6667% | 0.0207% |
| GunPointAgeSpan | 98.4177% | 98.7342% | 98.7342% | 0.3165% | 0.0003% |
| GunPointMaleVersusFemale | 99.6835% | 99.8101% | 99.6835% | 0.1266% | 0.0001% |
| GunPointOldVersusYoung | 99.0476% | 99.4921% | 100.0000% | 0.9524% | 0.0023% |
| Ham | 80.9524% | 79.4286% | 80.9524% | 1.5238% | 0.0077% |
| HandOutlines | 92.1622% | 92.8649% | 92.9730% | 0.8108% | 0.0019% |
| Haptics | 49.0260% | 50.9740% | 51.6234% | 2.5974% | 0.0183% |
| Herring | 65.6250% | 70.0000% | 67.1875% | 4.3750% | 0.0492% |
| HouseTwenty | 94.9580% | 95.4622% | 94.9580% | 0.5042% | 0.0008% |
| InlineSkate | 39.6364% | 40.5818% | 40.0000% | 0.9455% | 0.0023% |
| InsectEPGRegularTrain | 100.0000% | 100.0000% | 100.0000% | 0.0000% | 0.0000% |
| InsectEPGSmallTrain | 99.5984% | 99.5181% | 99.5984% | 0.0803% | 0.0000% |
| InsectWingbeatSound | 59.3434% | 59.7980% | 60.5051% | 1.1616% | 0.0034% |
| ItalyPowerDemand | 96.6958% | 96.8513% | 96.8902% | 0.1944% | 0.0001% |
| LargeKitchenAppliances | 84.5333% | 85.5467% | 84.8000% | 1.0133% | 0.0028% |
| Lightning2 | 86.8852% | 85.2459% | 85.2459% | 1.6393% | 0.0090% |
| Lightning7 | 82.1918% | 83.8356% | 84.9315% | 2.7397% | 0.0190% |
| Mallat | 93.2196% | 93.9275% | 95.8635% | 2.6439% | 0.0187% |
| Meat | 88.3333% | 92.6667% | 88.3333% | 4.3333% | 0.0626% |
| MedicalImages | 77.8947% | 78.1579% | 76.4474% | 1.7105% | 0.0085% |
| MelbournePedestrian | 91.9639% | 90.5945% | 90.8569% | 1.3694% | 0.0053% |
| MiddlePhalanxOutlineAgeGroup | 66.8831% | 66.7532% | 66.8831% | 0.1299% | 0.0001% |
| MiddlePhalanxOutlineCorrect | 85.2234% | 82.4055% | 81.4433% | 3.7801% | 0.0386% |
| MiddlePhalanxTW | 63.6364% | 64.2857% | 64.2857% | 0.6494% | 0.0014% |
| MixedShapesRegularTrain | 93.5258% | 92.9155% | 92.8660% | 0.6598% | 0.0014% |
| MixedShapesSmallTrain | 86.7216% | 85.7567% | 34.3918% | 52.3299% | 8.9628% |
| MoteStrain | 88.4984% | 91.8211% | 93.0511% | 4.5527% | 0.0555% |
| NonInvasiveFetalECGThorax1 | 93.1807% | 94.1374% | 93.5369% | 0.9567% | 0.0023% |
| NonInvasiveFetalECGThorax2 | 94.1985% | 94.0458% | 93.9949% | 0.2036% | 0.0001% |

**Table 7 continued from previous page**

| Dataset | $\sigma$=R | $\sigma$=R/2 | $\sigma$=R/3 | Difference | Variance |
|---|---|---|---|---|---|
| OliveOil | 73.3333% | 74.6667% | 80.0000% | 6.6667% | 0.1244% |
| OSULeaf | 75.6198% | 76.1983% | 74.7934% | 1.4050% | 0.0050% |
| PhalangesOutlinesCorrect | 83.6830% | 83.8462% | 84.7319% | 1.0490% | 0.0032% |
| Phoneme | 24.5781% | 26.3397% | 27.0042% | 2.4262% | 0.0157% |
| PickupGestureWiimoteZ | 74.0000% | 72.4000% | 78.0000% | 5.6000% | 0.0832% |
| PigAirwayPressure | 13.9423% | 23.9423% | 24.5192% | 10.5769% | 0.3537% |
| PigArtPressure | 53.3654% | 72.1154% | 74.0385% | 20.6731% | 1.3044% |
| PigCVP | 61.5385% | 69.0385% | 71.1538% | 9.6154% | 0.2553% |
| PLAID | 80.4469% | 81.5270% | 82.1229% | 1.6760% | 0.0072% |
| Plane | 100.0000% | 100.0000% | 100.0000% | 0.0000% | 0.0000% |
| PowerCons | 98.8889% | 98.2222% | 97.7778% | 1.1111% | 0.0031% |
| ProximalPhalanxOutlineAgeGroup | 87.8049% | 88.0976% | 87.8049% | 0.2927% | 0.0003% |
| ProximalPhalanxOutlineCorrect | 82.1306% | 83.3677% | 82.8179% | 1.2371% | 0.0038% |
| ProximalPhalanxTW | 82.9268% | 83.0244% | 83.9024% | 0.9756% | 0.0029% |
| RefrigerationDevices | 55.7333% | 56.1067% | 56.8000% | 1.0667% | 0.0029% |
| Rock | 74.0000% | 79.2000% | 74.0000% | 5.2000% | 0.0901% |
| ScreenType | 54.6667% | 52.2133% | 52.2667% | 2.4533% | 0.0196% |
| SemgHandGenderCh2 | 91.3333% | 90.8000% | 90.8333% | 0.5333% | 0.0009% |
| SemgHandMovementCh2 | 65.1111% | 64.6667% | 64.2222% | 0.8889% | 0.0020% |
| SemgHandSubjectCh2 | 86.4444% | 84.8889% | 84.6667% | 1.7778% | 0.0094% |
| ShakeGestureWiimoteZ | 94.0000% | 93.2000% | 92.0000% | 2.0000% | 0.0101% |
| ShapeletSim | 66.1111% | 67.1111% | 64.4444% | 2.6667% | 0.0181% |
| ShapesAll | 84.8333% | 85.3333% | 86.5000% | 1.6667% | 0.0073% |
| SmallKitchenAppliances | 74.9333% | 74.0267% | 72.8000% | 2.1333% | 0.0115% |
| SmoothSubspace | 98.6667% | 99.3333% | 100.0000% | 1.3333% | 0.0044% |
| SonyAIBORobotSurface1 | 92.1797% | 92.1797% | 92.1797% | 0.0000% | 0.0000% |
| SonyAIBORobotSurface2 | 90.4512% | 94.1868% | 93.4942% | 3.7356% | 0.0395% |
| StarLightCurves | 97.2317% | 97.5619% | 97.6688% | 0.4371% | 0.0005% |
| Strawberry | 97.0270% | 97.4054% | 97.5676% | 0.5405% | 0.0008% |
| SwedishLeaf | 92.9600% | 92.7360% | 91.8400% | 1.1200% | 0.0035% |
| Symbols | 88.6432% | 93.0452% | 95.6784% | 7.0352% | 0.1263% |
| SyntheticControl | 99.3333% | 99.9333% | 99.6667% | 0.6000% | 0.0009% |
| ToeSegmentation1 | 93.4211% | 95.2632% | 93.8596% | 1.8421% | 0.0093% |
| ToeSegmentation2 | 91.5385% | 92.4615% | 93.0769% | 1.5385% | 0.0060% |
| Trace | 100.0000% | 100.0000% | 100.0000% | 0.0000% | 0.0000% |
| TwoLeadECG | 99.6488% | 99.7191% | 99.7366% | 0.0878% | 0.0000% |
| TwoPatterns | 100.0000% | 100.0000% | 100.0000% | 0.0000% | 0.0000% |
| UMD | 98.6111% | 99.0278% | 99.3056% | 0.6944% | 0.0012% |
| UWaveGestureLibraryAll | 97.2641% | 96.9012% | 96.9570% | 0.3629% | 0.0004% |
| UWaveGestureLibraryX | 82.4400% | 82.5684% | 83.1658% | 0.7259% | 0.0015% |
| UWaveGestureLibraryY | 74.3439% | 74.3272% | 74.1485% | 0.1954% | 0.0001% |
| UWaveGestureLibraryZ | 75.6561% | 76.3261% | 76.8007% | 1.1446% | 0.0033% |
| Wafer | 99.6106% | 99.6820% | 99.7404% | 0.1298% | 0.0000% |
| Wine | 50.0000% | 76.6667% | 75.9259% | 26.6667% | 2.3064% |
| WordSynonyms | 70.0627% | 69.7179% | 68.6520% | 1.4107% | 0.0054% |
| Worms | 71.4286% | 64.6753% | 66.2338% | 6.7532% | 0.1250% |
| WormsTwoClass | 74.0260% | 78.9610% | 83.1169% | 9.0909% | 0.2071% |
| Yoga | 87.4333% | 86.8000% | 86.7000% | 0.7333% | 0.0016% |

