# OpenReview forum: "Time2Image: A Unified Image Representation Framework for Time Series Classification"
_ICLR.cc/2024/Conference — Submitted to ICLR 2024_

### Official Review · Reviewer_e5E4 · 2023-10-15

**Soundness:** 2 fair
**Presentation:** 3 good
**Contribution:** 1 poor
**Rating:** 5
**Confidence:** 4

**Summary:**

This paper proposes adaptive time series Gaussian mapping that converts a multivariate time series into 2D images with multiple channels. ATSGM gets a datapoint at a timestamp as a input and outputs sub-patches containing an image with a circle gaussian. Sub-patches are then merged to become a patch, the input for ViT architecture.

**Strengths:**

1. Novel idea to convert a time series into an image to exploit recent vision architecture ViT's generalization power.
2. The method is general to various time series domains.

**Weaknesses:**

1. No recent TSC baselines such as TimesNet[1], just using vision architectures although the task is time series classification.
2. No justification about why Gaussian should be used to generate subpatches.
3. Flaws in preprocessing. In preprocessing, resizing reduces the number of timestamps and can bring information loss. For example, a periodic pattern longer than the window size can vanish and a evolving pattern after a distribution shift can also be erased.


[1] Wu, Haixu, et al. "TimesNet: Temporal 2D-Variation Modeling for General Time Series Analysis." The Eleventh International Conference on Learning Representations. 2022.

**Questions:**

1. Is there any specific reason for the conversion from time series to gaussian image? It is straightforward to use segmented times series as the patches to ViT.

---

> ### Author Response · Authors · 2023-11-23
> **Official Response to Reviewer e5E4**
>
> **About weakness**:
>
> > No recent TSC baselines such as TimesNet[1], just using vision architectures, although the task is time series classification.
>
> **Answer**: Thank you for your suggestions on TimesNet, which is a strong and perfect model. We found that in this paper, the author adopted experiments on 10 UEA multivariate time series dataset. Due to time constraints, we couldn't conduct the TimeNet on all time series public dataset in a short time, even if we tried. Therefore, we recorded their results and compare it with the performance of Time2Image and we hope it can give some insight to better position our work.
>
> |        Dataset       | Time2Image |  TimesNet  |
> |:--------------------:|:----------:|:----------:|
> | EthanolConcentration |   28.67%   | **35.70%** |
> |     FaceDetection    |   67.79%   | **68.60%** |
> |      Handwriting     | **44.80%** |   32.10%   |
> |       Heartbeat      |   75.61%   | **78.00%** |
> |    JapaneseVowels    |   89.51%   | **98.40%** |
> |        PEMS-SF       |   84.62%   | **89.60%** |
> |  SelfRegulationSCP1  |   85.94%   | **91.80%** |
> |  SelfRegulationSCP2  | **60.22%** |   57.20%   |
> |  SpokenArabicDigits  | **99.41%** |   99.00%   |
> |  UWaveGestureLibrary | **89.31%** |   85.30%   |
>
> From the result, it can be seen that even though the performance of TimesNet outperform our method from these 10 datasets, we still have competitive performance, which is promising for time series image representation.
>
> > No justification about why Gaussian should be used to generate subpatches.
>
> **Answer**: Thank you for your questions. We think this question is similar to the motivation or intuition of Gaussian mapping.  We think answers to overall response can be Answers to general response **Inspired by 2D keypoint detection task** in computer vision, we noticed that Gaussian distributions are often used to model uncertainty or confidence in locations. In addition, a Gaussian distribution is a smooth distribution; that is to say, since time series data usually have noise and fluctuations, we think modeling it through a Gaussian distribution can smooth the data to a certain extent and make the model more robust. Therefore, introducing this idea into the representation of time series data by modeling the uncertainty of each channel time point and forming a sub-patch through Adaptive Time Series Gaussian Mapping(ATSGM) we proposed, can help the model better understand the importance of each time point in the time series data. At the same time, the relationship and local information between different channels in MTS data can also be obtained in order to improve the model's robustness and generalization ability. Each subpatch i ​​into the ViT model in chronological order, so that the model can effectively capture long-term information.
>
> > Flaws in preprocessing. In preprocessing, resizing reduces the number of timestamps and can bring information loss. For example, a periodic pattern longer than the window size can vanish, and an evolving pattern after a distribution shift can also be erased.
>
> **Answer**: We really appreciate your insightful and valuable concern. We use resizing in our method for two reasons. First, it helps handle time series of different lengths. Second, it aligns with the input requirements of our chosen model, ViT-B/16.  We need each subpatch to represent a Gaussian mapping of a specific time across different channels. We picked cubic interpolation during resizing since it is smooth and maintains overall trends compared to other interpolation methods. However, we acknowledge there's still some information loss with significant resizing. That is why we tested our approach on all public datasets to get an overall performance measure. And from the result, it shows that our proposed algorithm is suitable for most time series data with a certain length, but for the case with extremely long time series, our method performs not that well compared to state-of-the-art models, as the reviewer pointed out. We appreciate this feedback, and it guides our future work. The adjustment of the interpolation method for pre-processing or the modification on image classification algorithm could be two directions for further work. We hope this could address your questions.
>
> **About Question**:
> > Is there any specific reason for the conversion from time series to gaussian image? It is straightforward to use segmented times series as the patches to ViT.
>
> **Answer**: We appreciate the reviewer's question, which we think is still similar to the motivation and intuition of gaussian mapping. Please see our answers to Weakness #2. Thanks.
>
> Thank you again for your insightful questions and valuable advice. If you find that our idea is good and our response addresses your concerns to some extent, could you please consider increasing your rating score for our paper? Your consideration is highly appreciated.

---

### Official Review · Reviewer_HAQG · 2023-10-23

**Soundness:** 2 fair
**Presentation:** 2 fair
**Contribution:** 3 good
**Rating:** 5
**Confidence:** 4

**Summary:**

This paper proposes a new approach to time series classification (TSC). The method, Time2Image, involves producing image-based representations of time series, and then classifying the image representations using a vision transformer (ViT). The encoding process uses a  novel technique called Adaptive Time Series Gaussian Mapping (TSGM). Time2Image is able to process both univariate and multivariate time series, and can handle time series of unequal lengths. The approach is evaluated against SOTA methods on 158 datasets from the UCR/UEA collection.

**Strengths:**

**Originality**
O1. The novelty of the methods lies in the time series to image encoding process (ATSGM), which is a novel way of encoding time series.

**Quality**
Q1. Compared to existing time series image representation methods (GAF, MTF, and RP) using the same classifier (ViT), Time2Image is shown to have improved performance on UTS datasets (Figure 3).
Q2. Compared to two baseline methods (FCN and ResNet), Time2Image is shown to have the best performance on UTS (Table 1 and Figure 3).
Q3. For the MTS datasets, Time2Image wins on the most datasets when compared to five baselines (ResNet, ROCKET, CIF, HIVE-COTE, and InceptionTime).
Q4. The results are shown to generalise across different univariate TSC domains (Table 2).

**Clarity**
C1. I appreciate the inclusion of Algorithm 1 to aid understanding and clarity of the method.

**Significance**
S1. Time series image representation methods are well motivated, as powerful methods for computer vision can then be utilised. As existing approaches are behind SOTA, improving on them helps move the general family of methods forward.

**Weaknesses:**

**Originality**
O1. Further comparison to existing time series to image encoding methods is required, demonstrating the benefits of ATSGM. e.g. evaluation on MTS.
O2. The classification method (ViT) is unchanged from its existing design, so originality only appears in the image encoding process.

**Quality**
Q1. **My biggest criticism of the work is the selection of SOTA baselines to compare to.** A recent study [1] identified HIVE-COTEv2 and Hydra+MultiROCKET as the new SOTA for univariate time series. These methods would be better comparisons for UTS than FCN and ResNet, and help better position the work.
Q2. I am unsure why only FCN and ResNet were used for the UTS evaluation. The other SOTA baselines that are used for MTS can be applied to UTS as well, but are not used.

**Clarity**
C1.  It needs to be made more clear that all time series are resized to fixed length L (which is set to 196 for the experiments). I could not see L explicitly defined.
C2. The link in footnote 1 does not work (404 not found). Visual examples of "circle packing in a square" would aid understanding.
C3. Figures 1 and 2 need improvement. Additional information in the caption and further explanation of the patch -> matrix reshaping (step 11 in Algorithm 1) would be helpful.
C4. I believe the ECG results are incorrect in Table 2. ResNet has an accuracy of 94.98%, while Time2Image has an accuracy of 94.67%. However, Time2Image is highlighted as the best score.

**Signifiance**
S1. Due to the choice of baselines, it is difficult to determine how significant the development of Time2Image is.
S2. Many of these datasets are imbalanced, so only evaluating with accuracy can be misleading. A further evaluation using balanced accuracy, AUROC, etc. would be beneficial.
S3. While grouping by domain does show generalisation, it would be more insightful to analyse by dataset properties, e.g. time series length, number of classes etc. This would help identify where the method is strong and where it is weak.
S4. In addition to providing the critical difference diagram and number of wins, it would be helpful to see the average accuracy over all datasets.

[1] Middlehurst, Matthew, Patrick Schäfer, and Anthony Bagnall. "Bake off redux: a review and experimental evaluation of recent time series classification algorithms." arXiv preprint arXiv:2304.13029 (2023).

**Questions:**

Q1. Why were ROCKET, CIF, HIVE-COTE, and InceptionTime not evaluated for UTS?
Q2. Other than improved performance, what are the benefits of ATSGM over existing image encoding methods? E.g. are the other methods able to deal with multivariate time series?

---

> ### Author Response · Authors · 2023-11-23
> **Official Response to Reviewer HAQG on Weaknesses 1/3**
>
> We greatly appreciate your thoughtful questions and valuable suggestions, which have significantly enriched the discussion around our work. Please find the answers below.
>
> **About weakness**
> > O1. Further comparison to existing time series to image encoding methods is required, demonstrating the benefits of ATSGM. e.g. evaluation on MTS.
>
> **Answer**: We sincerely appreciate your thorough review of our work. We would like to clarify that the reason we did not compare our method with existing time series image encoding methods on Multivariate Time Series (MTS) is due to the inherent challenge of extending current time series modeling methods to handle multivariate cases. As for now, there is a lack of methods specifically designed for multivariate time series image representation, and our proposed ATSGM is the first method to address this by providing a novel approach for modeling multivariate time series as images. We hope it could become the first baseline for comparison in the future.
>
> > O2. The classification method (ViT) is unchanged from its existing design, so originality only appears in the image encoding process.
>
> **Answer**:Thanks for your insightful observation. Indeed, the core classification method (ViT) remains unchanged in its original design, and the primary innovation in our work lies in the novel image encoding process—ATSGM. From the overall response, we mentioned 2 research gap on time series image representation for classification, which we will quote here: (1) Before this work, there are no research working on multivariate time series (MTS) image representations since it is a challenging task. (2) Even though there are some image representation approaches for univariate time series, there are significant performance gaps to make a comparison of current state-of-the-art (SOTA) models in classification tasks. By introducing the Gaussian projection and utilizing sub-patches constructed from each channel's time points, we aim to provide a unique and effective representation for both univariate and multivariate time series data. The use of Gaussian projection allows for flexible modeling of the non-linear structures present in time series, contributing to improved feature retention and robustness. We believe that the novelty and contributions of our work primarily stem from this encoding process, enabling a distinctive approach to time series representation. Hope this answer can address your concern.
>
> > Q1. My biggest criticism of the work is the selection of SOTA baselines to compare to. A recent study [1] identified HIVE-COTEv2 and Hydra+MultiROCKET as the new SOTA for univariate time series. These methods would be better comparisons for UTS than FCN and ResNet, and help better position the work.
>
> **Answer**: We appreciate your attention to the field of Time Series Classification (TSC) and thank you for highlighting the recent work published in April [1]. However, what we want to say is that the initial goal of our study was to demonstrate the immense potential of time series image representation, showcasing the substantial performance gap between existing image representation methods and state-of-the-art (SOTA) methods in the context of image classification. Additionally, we aimed to explore the effectiveness of combining image representation with deep learning methods, bridging the gap between time series and computer vision. Therefore, in baseline model selection process, we only consider the deep learning-based time series classification algorithm. While other SOTA baselines (e.g.HIVE-COTEv2 and Hydra+MultiROCKET) all belong to the hybrid (ensemble) model according to the literature you just mentioned, and we think our selection in the comparative experiments should be driven by the desire for a fair comparison between models and a focus on showcasing the performance of our proposed method within deep learning frameworks.
>
> > Q2. I am unsure why only FCN and ResNet were used for the UTS evaluation. The other SOTA baselines that are used for MTS can be applied to UTS as well, but are not used.
>
> **Answer**: Thanks for your question. This concern is similar to Q1. Please find our answers to Q1 and our overall response. Any suggestions and comments are welcome if there's any question.

---

> ### Author Response · Authors · 2023-11-23
> **Official Response to Reviewer HAQG on Weaknesses 2/3**
>
> > C1. It needs to be made more clear that all time series are resized to fixed length L (which is set to 196 for the experiments). I could not see L explicitly defined.
>
> **Answer**: Thanks once again for your valuable suggestions. We also apologize for any lack of clarity in defining this aspect. As you correctly pointed out, for our experiments, we chose L to be 196. The primary reason for this choice is the subsequent use of the ViT-B/16 classification model. In response to your feedback, we have further elaborated on this aspect in our revised version, which will be uploaded later. We appreciate your diligence in reviewing our work and believe that these revisions will contribute to a more comprehensive understanding of our experimental setup.
>
> > C2. The link in footnote 1 does not work (404 not found). Visual examples of "circle packing in a square" would aid understanding.
>
> **Answer**: We appreciate your attention to detail and thank you for bringing this to our attention. We apologize for the broken link in footnote 1, which has now been rectified in the revised version of the manuscript. The correct link is now: http://hydra.nat.uni-magdeburg.de/packing/csq/csq.html. We hope this website can help. Further questions or suggestions would also be appreciated.
>
> > C3. Figures 1 and 2 need improvement. Additional information in the caption and further explanation of the patch -> matrix reshaping (step 11 in Algorithm 1) would be helpful.
>
> **Answer**: We appreciate your feedback and will certainly work on improving the clarity of Figures 1 and 2. In the revised manuscript, we will provide more detailed information in the captions to offer better context for the figures. Additionally, we recognize the importance of explaining the patch -> matrix reshaping process (step 11 in Algorithm 1). Please find them in the captions of Figure 1 and 2 in our revised manuscript. And thanks for your suggestions for a better understanding of our framework.
>
> > C4. I believe the ECG results are incorrect in Table 2. ResNet has an accuracy of 94.98%, while Time2Image has an accuracy of 94.67%. However, Time2Image is highlighted as the best score.
>
> **Answer**: Thank you for bringing this to our attention, and we apologize for any confusion. We will rectify this in the revised version and ensure that the correct information is presented. Please find the revised version later. Thanks for making our work better. However, it's crucial to note that despite this oversight, the correct ranking and result from the analysis remain the same, which shows the inherent generalization ability of Time2Image.
>
> > S1. Due to the choice of baselines, it is difficult to determine how significant the development of Time2Image is.
>
> **Answer**: We appreciate your concern regarding the choice of baselines in our study. The primary objective of our work was to explore the effectiveness of the proposed Time2Image representation for time series classification, emphasizing its adaptability to both univariate and multivariate scenarios. Therefore, in our opinion, the focus should be more on the comparison between different time series image representations and the current deep learning image classification architecture. More answers can be found in Q1 and the overall response. But please note that we are open to further discussions and recommendations on baseline model selection.

---

> ### Author Response · Authors · 2023-11-23
> **Official Response to Reviewer HAQG on Weaknesses 3/3**
>
> > S2. Many of these datasets are imbalanced, so only evaluating with accuracy can be misleading. A further evaluation using balanced accuracy, AUROC, etc. would be beneficial.
>
> **Answer**: Thank you very much for your suggestion. For this reason, we conducted experiments on the performance of our model, obtained a balanced accuracy value, and compared it with the accuracy we disclosed before. We selected 14 UTS data and 10 MTS data with various lengths, classes, domains, etc. due to time constraints. The results can be seen below. From the results, we can see that by calculating the balanced accuracy, we can find that in some datasets, balanced accuracy performs better than accuracy on most datasets, which also shows that our method can solve the problem of imbalanced dataset well.
>
> | Dataset                    | Balanced Accuracy | Accuracy |
> |----------------------------|-------------------|----------|
> | ECGFiveDays                | **93.17%**         | 91.87%   |
> | ItalyPowerDemand           | 96.79%            | **96.85%**   |
> | ECG200                     | 92.64%            |**93.40%**   |
> | GestureMidAirD1            | 71.72%            | **72.77%**  |
> | EOGHorizontalSignal        | **63.74%**           | 60.61%   |
> | EOGVerticalSignal          |**57.87%**            | 55.36%   |
> | ECG5000                    | 92.23%            | **94.82%**   |
> | FordA                      |**94.05%**           | 93.98%   |
> | FordB                      | **81.13%**          | 80.86%   |
> | PickupGestureWiimoteZ      | **78.47%**            | 72.40%   |
> | ShakeGestureWiimoteZ       | **96.67%**             | 93.20%   |
> | ItalyPowerDemand           | 96.79%            | **96.85%**  |
> | NonInvasiveFetalECGThorax1 | **94.59%**           | 94.14%   |
> | NonInvasiveFetalECGThorax2 | **94.05%**            | **94.05%**   |
>
> | Dataset              | Balanced Accuracy | Accuracy |
> |----------------------|-------------------|----------|
> | EthanolConcentration | **62.64%**            | 28.67%   |
> |     FaceDetection    | 67.40%            | **67.79%**   |
> |      Handwriting     | **49.85%**            | 44.80%   |
> |       Heartbeat      | 74.56%            | **75.61%**   |
> |    JapaneseVowels    | **91.38%**            | 89.51%   |
> |        PEMS-SF       | **87.79%**            | 84.62%   |
> |  SelfRegulationSCP1  | **87.03%**            | 85.94%   |
> |  SelfRegulationSCP2  | **62.59%**            | 60.22%   |
> |  SpokenArabicDigits  | **99.45%**           | 99.41%   |
> |  UWaveGestureLibrary | **90.59%**            | 89.31%   |
>
> > S3. While grouping by domain does show generalisation, it would be more insightful to analyse by dataset properties, e.g. time series length, number of classes etc. This would help identify where the method is strong and where it is weak.
>
> **Answer**: Thank you for your suggestions. Actually, reviewer mdnq also asked a similar question, which requires an in-depth analysis from a data perspective to help understand the bounds and weaknesses of the model. Here we paste the answers and share our findings with you from previous answers:
>
> **(1) The number of channels in MTS will have an important impact on the performance**; that is to say, our method might have limited performance when the number of MTS is large. Since the size of the sub-patch is fixed and we regard data on each dataset from different channels as "equal circle in the square" problem, that means the larger the channel, the smaller the place for representation of each channel (see DuckDuckGeese with 1345 channels, PEMS_SF with 963 channels,etc.).
>
> **(2) The length of the time series will have an impact on the performance**; that is to say, the smaller the difference between predefined sizes L, the better the performance.The reason is mainly because of the resize we use in preprocessing. In this work, we set size L to 196. Even though resizing can deal with unequal length on MTS, it is recommended to find the suitable size based on the original length of the dataset in the future. For instance, when the length of the time series is too long, there will be a large information loss, which might have a negative impact on the performance (see EigenWorms with 17984, CinCECGTorso with 1639, and HandOutlines with 2709).I hope these findings will address your concern. Any further suggestions and comments are still welcome.
>
> > S4. In addition to providing the critical difference diagram and number of wins, it would be helpful to see the average accuracy over all datasets.
>
> **Answer**: Thank you for your suggestions. We have added the average accuracy of each method to the Table 5 and 6 in the appendix. Through analysis, it can be found that our method still has the best performance in average accuracy in MTS an has a slightly gap with InceptionTime.

---

> ### Author Response · Authors · 2023-11-23
> **Official Response to Reviewer HAQG on Questions**
>
> > Q1. Why were ROCKET, CIF, HIVE-COTE, and InceptionTime not evaluated for UTS?
>
> **Answer**: Thank you for this question. We added InceptionTime as one of our baseline for UTS since recent paper categorized this method as a deep learning-based classification algorithm, and the performance can be found in our revised manuscript. However, since ROCKET, CIF, and HIVE-COTE all belong to ensemble algorithms, which is not what we are focused on. Further explanation can be found in the answers to Weakness Q1 and the overall response. Thank you for understanding.
>
>
> >Q2. Other than improved performance, what are the benefits of ATSGM over existing image encoding methods? E.g. are the other methods able to deal with multivariate time series?
>
> **Answer**: Thank you so much for this insightful question. Actually, there is no research working on multivariate time series image representation, so our work is the first to give a possible solution for image representation. In addition to the enhanced performance, ATSGM holds distinct advantages over existing image encoding methods, particularly in its ability to effectively handle multivariate time series (MTS) data. The Gaussian mapping employed by ATSGM allows for a better representation of MTS, capturing relationships between different channels or variables within the time series. This is a crucial feature, as many traditional image encoding methods may struggle to adequately model the intricate dependencies present in MTS data. Therefore, ATSGM's specialization in encoding multivariate time series contributes a unique and valuable dimension to the field of image representation for time series classification.
>
>
> Thank you again for your insightful questions and valuable advice. If you have any more questions, we're more than willing to continue the discussion. If you find that our response addresses your concerns, could you please consider increasing your rating score for our paper? Your consideration is highly appreciated.

---

> ### Comment · Reviewer_HAQG · 2023-11-23
>
> Thank you very much for your extensive response to my review.
>
> As the proposed method is the first to perform multivariate time series image representation, the work now has more weight in terms of novelty. I suggest authors make this as clear as possible in the paper itself (apologies if I missed this on my first read through).
>
> I appreciate the need for a fair comparison with deep learning methods. However, excluding other SOTA methods because they come from another family does not seem a sufficient reason to not include them. In my opinion, the results of other methods should be included at least to provide context and demonstrate the gains in performance of the new method (closing the gap to SOTA), even if these methods are not necessarily a fair comparison as they are ensemble approaches.
>
> I'm not sure if this is a problem on my end, but I cannot see the changes of the revised version. Has it been submitted to OpenReview?
>
> From the rebuttal, specifically with the clarification regarding novelty, I have increased my review score from 3 to 5. However, I am unwilling to increase it further without seeing the revised version of the paper. Again, apologies if this is an issue on my end but I cannot see the revised version.

---

### Official Review · Reviewer_soQ5 · 2023-10-30

**Soundness:** 3 good
**Presentation:** 3 good
**Contribution:** 2 fair
**Rating:** 5
**Confidence:** 4

**Summary:**

The paper introduces a method to transform time series data into images, subsequently employing an image classifier to classify the time series data. This image representation derives from calculating the Gaussian distribution for each subsequence. The primary objective of the proposed method is to enhance time series classification.

**Strengths:**

Paper is very well written and structured.

**Weaknesses:**

Missing some related works:

A recent survey, as seen in [https://arxiv.org/pdf/2304.13029.pdf], found that InceptionTime, detailed at [https://link.springer.com/article/10.1007/s10618-020-00710-y], achieves higher accuracy than ResNet. A comparison of the proposed method with the InceptionTime model could provide further insights into the proposed method performance.

Furthermore, the paper omits some pertinent related work, such as methods based on the Fast Fourier Transform (FFT) and the Continuous Wavelet Transform (CWT).
1-Meintjes, A., Lowe, A., & Legget, M. (2018, July). Fundamental heart sound classification using the continuous wavelet transform and convolutional neural networks. In 2018 40th annual international conference of the IEEE engineering in medicine and biology society (EMBC) (pp. 409-412). IEEE.
2- Hatami, N., Gavet, Y., & Debayle, J. (2018, April). Classification of time-series images using deep convolutional neural networks. In Tenth international conference on machine vision (ICMV 2017) (Vol. 10696, pp. 242-249). SPIE.

**Questions:**

The core goal of transforming time series data into images is to boost accuracy. Although the authors demonstrate that their method outperforms other networks that utilize raw time series data, such as the Fully Connected Network (FCN) and ResNet, this superiority isn't primarily due to the image transformation. Instead, it's attributed to the disparity in network size. For instance, the FCN comprises 396,550 parameters, and ResNet contains 580,486 parameters. In contrast, the network featured in this paper, VIT-16, encompasses a staggering 86,859,496 parameters. Despite its immense size, VIT-16 significantly surpasses the other networks. However, given the smaller size of the other networks, they still yield commendable results. One must question the computational feasibility of transforming time series data into images and then training a model on it. Wouldn't a larger ResNet or FCN model potentially produce comparable results?

---

> ### Author Response · Authors · 2023-11-23
> **Official Response to Reviewer soQ5 on Weakness**
>
> Thank you for the insightful comments on our work. Please find our answers below.
>
> **About weakness**
> > Missing some related works: A recent survey, as seen in [https://arxiv.org/pdf/2304.13029.pdf], found that InceptionTime, detailed at [https://link.springer.com/article/10.1007/s10618-020-00710-y], achieves higher accuracy than ResNet. A comparison of the proposed method with the InceptionTime model could provide further insights into the proposed method's performance.
>
> **Answer**: Thank you for highlighting InceptionTime as a competitive baseline. In our initial manuscript, our primary focus was to demonstrate the effectiveness of time series image representation through deep learning-based classification. We chose FCN and ResNet as baselines based on a comprehensive review of deep learning models for classification [1], considering them as SOTA models for our specific task. We were concerned that including InceptionTime, designed as a neural network ensemble model for UTS, might introduce bias and result in an unfair comparison. However, we appreciate your valuable suggestion and found that in a recent paper you mentioned, the author also mentioned InceptionTime as one of the deep learning-based methods, and in response, we have included the performance comparison with InceptionTime on UTS in our Appendix Table 5 and update Figure 3 in revised manuscript, please find the CD diagram including InceptionTime. It is worth noting that through CD diagram, it can be found that even with this additional comparison, our proposed Time2Image method continues to exhibit superiority on both UTS and MTS datasets.
> [1] Ismail et al. Deep learning for time series classification:a review
>
>
> > Furthermore, the paper omits some pertinent related work, such as methods based on the Fast Fourier Transform (FFT) and the Continuous Wavelet Transform (CWT). 1-Meintjes, A., Lowe, A., & Legget, M. (2018, July). Fundamental heart sound classification using the continuous wavelet transform and convolutional neural networks. In 2018 40th annual international conference of the IEEE engineering in medicine and biology society (EMBC) (pp. 409-412). IEEE. 2- Hatami, N., Gavet, Y., & Debayle, J. (2018, April). Classification of time-series images using deep convolutional neural networks. In Tenth international conference on machine vision (ICMV 2017) (Vol. 10696, pp. 242-249). SPIE.
>
> Answer: Thank you for bringing to our attention the two relevant papers on time series classification. After reviewing these works, we have made the following adjustments to the Related Work section:
> - The first paper introduces another time series encoding method, which we believe falls under the category of decomposition methods within the broader transformation methods. We have incorporated this information into the first paragraph of Section 2.1.
> - The second paper presents RP+CNN for univariate time series classification, representing one of the time series image representations followed by classification algorithms. This content has been included in the second paragraph when introducing time series image encoding methods.
> We appreciate your valuable suggestions, and please find these two papers in the related work section of our revised manuscript.

---

> ### Author Response · Authors · 2023-11-23
> **Official Response to Reviewer soQ5 on Questions**
>
> **About questions**
> > The core goal of transforming time series data into images is to boost accuracy. Although the authors demonstrate that their method outperforms other networks that utilize raw time series data, such as the Fully Connected Network (FCN) and ResNet, this superiority isn't primarily due to the image transformation. Instead, it's attributed to the disparity in network size. For instance, the FCN comprises 396,550 parameters, and ResNet contains 580,486 parameters. In contrast, the network featured in this paper, VIT-16, encompasses a staggering 86,859,496 parameters. Despite its immense size, VIT-16 significantly surpasses the other networks. However, given the smaller size of the other networks, they still yield commendable results. One must question the computational feasibility of transforming time series data into images and then training a model on it. Wouldn't a larger ResNet or FCN model potentially produce comparable results?
>
> **Answer**: Thank you for bringing this to our attention, and we find your suggestion intriguing for further exploration. Over the past few days, we conducted additional experiments to test the performance on larger ResNet, and now we would like to share the results with you. The experiment was conducted through all UEA Multivariate time series dataset. Specifically, we utilized ResNeXt101-32x8d, which encompasses **88,791,336** parameters, sharing a similar scale parameter with ViT-B/16.
>
> The accuracy of ResNet-50, ResNeXt101-32x8d, and our proposed Time2Image on all UEA MTS datasets is presented below.
> | Dataset                   | Time2Image  | ResNeXt101-32x8d  | ResNet     |
> |---------------------------|-------------|------------|------------|
> | ArticularyWordRecognition | 97.80%      | 96.67%     | **98.26%** |
> |     AtrialFibrillation    | 45.33%      | **60.00%** | 36.22%     |
> |        BasicMotions       | 99.00%  | 80.00%     | **100.00%**    |
> |   CharacterTrajectories   | **99.50%**  | 99.30%     | -          |
> |          Cricket          | **100.00%** | 91.67%     | 99.40%     |
> |       DuckDuckGeese       | 47.20%      | 44.00%     | **63.20%** |
> |           ERing           | **89.93%**  | 80.74%     | 45.45%     |
> |         EigenWorms        | 79.24%      | 42.75%     | **99.18%** |
> |          Epilepsy         | **96.09%**  | 84.06%     | 28.66%     |
> |    EthanolConcentration   | 28.67%      | 25.10%     | **87.19%** |
> |       FaceDetection       | **67.79%**  | 50.88%     | 62.97%     |
> |      FingerMovements      | **58.00%**  | 53.00%     | 54.70%     |
> |   HandMovementDirection   | **61.89%**  | 40.54%     | 35.32%     |
> |        Handwriting        | 44.80%      | 24.35%     | **59.78%** |
> |         Heartbeat         | **75.61%**  | 72.20%     | 63.89%     |
> |       InsectWingbeat      | 47.60%      | 41.64%     | -          |
> |       JapaneseVowels      | **89.51%**  | 82.70%     | -          |
> |            LSST           | 65.24%      | 62.49%     | **94.11%** |
> |           Libras          | **86.44%**  | 82.22%     | 42.94%     |
> |        MotorImagery       | **63.60%**  | 53.00%     | 49.77%     |
> |           NATOPS          | 86.33%      | 77.22%     | **97.11%** |
> |          PEMS-SF          | 84.62%      | 79.77%     | **99.64%** |
> |         PenDigits         | **99.11%**  | 99.00%     | 81.95%     |
> |       PhonemeSpectra      | **24.94%**  | 23.11%     | 30.86%     |
> |        RacketSports       | 89.87%      | 79.61%     | **94.23%** |
> |     SelfRegulationSCP1    | 85.94%      | **86.69%** | 76.11%     |
> |     SelfRegulationSCP2    | **60.22%**  | 54.44%     | 50.24%     |
> |     SpokenArabicDigits    | 99.41%      | **99.77%** | -          |
> |       StandWalkJump       | **66.67%**  | 53.33%     | 30.89%     |
> |    UWaveGestureLibrary    | 89.31%      | **89.38%** | 88.35%     |
>
> From the result, we can see that despite the increased parameterization in ResNeXt101-32x8d, which has **88,791,336 parameters** compared to the **86,859,496 parameters** in ViT-B, our Time2Image method consistently outperforms on 15 out of 32 datasets. This superior performance, even with the significantly larger parameter count of ResNeXt101-32x8d relative to the standard ResNet-50, suggests that increased model size does not linearly correlate with improved results. Importantly, our findings highlight the practical viability of Time2Image, especially in contexts where accuracy is paramount, despite potential increases in computational time. Thank you for inspiring us to conduct this insightful investigation.
>
> We're grateful for your thoughtful questions and helpful advice. Please feel free to reach out if you have any additional questions. If you feel that our response adequately addresses your concerns, would you like to consider raising your rating score for our paper? Thank you for your consideration!

---

### Official Review · Reviewer_mdnq · 2023-11-01

**Soundness:** 3 good
**Presentation:** 3 good
**Contribution:** 3 good
**Rating:** 6
**Confidence:** 4

**Summary:**

This work presents a method for time series feature representation that enables the use of transformers on multivariate time series classification problems in a new, performant way. This representation, Adaptive Time Series Gaussian Mapping frames the representation as a packing problem for Gaussian distributions, where all channels in a multivariate time series are represented by an individual Gaussian distribution projected onto a 16x16 patch. A single patch represents each time step in the sequence and a matrix of all the patches represents the entire sequence. Classification is then completed on the final representation with evaluation on standard benchmarks showing improvement over current methods on univariate classification and matching performance on multivariate classification.

**Strengths:**

Thanks to the authors for their submission: it contains useful research that shows good research practices while explaining an interesting and novel idea within multivariate time series classification. The results of this work will be informative to other researchers and are significant in improving our understanding of applying deep learning methods within time series work - a field where such applications have proved difficult.

This paper adds to a small but growing literature around the use of Transformer-based models on multivariate time series classification. While previous large-scale benchmarks have shown that the best deep neural network-based methods use convolutional methods (ResNet, InceptionTime), this approach shows how to combine time series with Vision Transformers, which have shown improvements over ResNet in computer vision. The ATSGM framework which packs each time stamp into a patch representation that can then be fed into ViT, provides a reasonable way to represent the time series.

Evaluation of the method against other leading approaches is thorough, using the standard UCI univariate and multivariate benchmarks. Conducting full benchmarking across all the tasks is an accomplishment which few papers in time series choose to complete and the authors should be lauded for this! The evidence provided shows that ATSGM+ViT outperforms existing methods in the univariate setting and has matching performance with the best methods in the multivariate setting.

Finally the construction of this paper was strong. The writing was well put-together with helpful diagrams elucidating the more difficult parts of the provided methods. It was easy to read and to the point.

**Weaknesses:**

There are two key weakness that I see in this paper:

One is the lack of comparison with other encoding and representation methods combined with transformers. There is previous work [1, 2] that uses normalized and learnable linearly projected representations of the multivariate time series with learnable positional encodings. Additional ablation studies that separate out the normalization and projection methods and compare against additional encoding types could help shine a light on the specific benefits of the ATSGM framework. One concern I have is that the improvement may be primarily due to ViT using positional embeddings and a linear projection of the flattened patches rather than due to the Gaussian projection itself and the Gaussian method may be adding more unnecessary complexity.

Second is the lack of motivation for the main method. While it is a novel method which performs well in the standardized benchmarks, it is not clear what the bounds or weaknesses of method might be. I have suggested a number in the questions below. In short: how does the Gaussian representation restrict the information available as time series scale up in size? How might the method be flexible enough to accommodate different data needs?

[1] “TST”  https://arxiv.org/pdf/2010.02803.pdf
[2] “CropTransformer” https://arxiv.org/pdf/1905.11893.pdf


Additional Nits:
- Page 5: the link to the best packings of equal circles in a square returns 404. The correct link should include a ‘#’ before ‘Results’.
- Algorithm 1 does not define L anywhere in the paper though I believe it is simply equal to 196.
- Table 1: Incorrect based on the results of Table 6. In Table 6 ResNet shows 4 wins instead of 3, Time2Image shows 11 wins instead of 13, and Time2Image+ResNet shows 2 wins which should not be conflated with Time2Image+ViT in this table.
- Figure 4 should include InceptionTime

**Questions:**

These questions will help clarify my understanding of the paper. Some of these could benefit from additional analysis in the paper itself:

1/ What are the author’s intuitions for why Time2Image underperforms the regular ResNet? Is there something about the Gaussian representation that is particularly suited to Transformers over Convulational approaches?

2/ The results of InceptionTime seem to worse in this evaluation compared with the results of the Time Series Benchmark and Redux. Is this because T2I is better at the same categories as inception time? While the paper explains the generalization ability of T2I across domains it does not shine any light on where (if any) the gains may come from.

3/ Are the authors concerned that the method  for selecting the standard deviation parameter could allow overfitting on the benchmark datasets? Were there any held-out datasets for parameter selection that could be used for evaluation?

4/ Given the restrictions of the algorithm, in that finding a packing for the gaussian distributions in the patches will reduce the resolution of the final sample as the number of channels increases, would it make sense to evaluate the performance of the model as the number of channels increases? From quick observation on the datasets with high number of channels (Crop-7200, phalangesOutlinesCorrect-1800, Ford*-3600, ElectricDevices-8926, NonInvasive*-1800, DuckDuckGeese-1345) it seems that in this collection the T2I win-rate is lower (~50%). A similar analysis would be on sequence length. EigenWorms contains the longest sequences by an order of magnitude in the MTS set and Time2Image performance is much worse than ResNet or InceptionTime. Helping understand these performance bounds would be very useful for practical applications and understanding the generalization from the test benchmark to other evaluations. This could also help inform how the additional parameters for series normalization and patch size can be set.

---

> ### Author Response · Authors · 2023-11-23
> **Official Response to Reviewer mdnq on weakness 1/2**
>
> Thank you your thoughtful and constructive feedback on our manuscript. We appreciate the time and effort you have dedicated to reviewing our work. Below, we address your concerns and provide additional insights into our methodology.
>
> >  W1: lack of comparison with other encoding and representation methods, i.e. w/o normalization, and position embeddings.
>
> __Answer__:
> Thank you for your detailed review and valuable suggestions. In order to fully respond to your above requests, we have conducted some experiments. **First**, we conducted a normalized ablation experiment on 10 selected MTS datasets and 10 selected UTS datasets according to different length, class and domains during preprocessing for our proposed method to study the impact of normalization on the classification performance. The **accuracy** of each dataset and the **epochs** required to achieve the best performance can be seen as follows:
> |    UCR UTS Datasets   |  Time2Image  | Time2Image w.o. Normalization |
> |:---------------------:|:------------:|:-----------------------------:|
> |      ECGFiveDays      |  91.87% (89) |          92.57% (122)         |
> | PickupGestureWiimoteZ | 72.40% (102) |          72.00% (130)         |
> |  ShakeGestureWiimoteZ |  93.20% (96) |          95.99% (139)         |
> |    ItalyPowerDemand   |  96.85% (75) |          96.79% (96)          |
> |         ECG200        |  93.40% (90) |           93.00%(95)          |
> |    GestureMidAirD1    | 72.77% (102) |          64.62% (127)         |
> |  EOGHorizontalSignal  |  60.61% (33) |          59.94% (38)          |
> |   EOGVerticalSignal   |  55.36% (40) |          54.41% (47)          |
> |        ECG5000        |  94.82% (62) |          94.79% (75)          |
> |         FordA         |  93.98% (42) |           94.02%(34)          |
>
> | UEA MTS Datasets     | Time2Image   | Time2Image w.o. Normalization |
> |----------------------|--------------|-------------------------------|
> | EthanolConcentration | 28.67% (64)  | 35.36% (99)                   |
> | FaceDetection        | 67.79% (12)  | 66.91% (12)                   |
> | Handwriting          | 44.80% (123) | 42.59% (184)                  |
> | Heartbeat            | 75.61% (44)  | 73.66% (77)                   |
> | JapaneseVowels       | 89.51% (33)  | 87.84% (51)                   |
> | PEMS-SF              | 84.62% (92)  | 86.71% (181)                  |
> | SelfRegulationSCP1   | 85.94% (38)  | 90.44% (32)                   |
> | SelfRegulationSCP2   | 60.22% (68)  | 56.67%(132)                   |
> | SpokenArabicDigits   | 99.41% (30)  | 99.41% (36)                   |
> | UWaveGestureLibrary  | 89.31% (55)  | 90.94% (78)                   |
>
> According to the data presented in the table, we observe that the impact of normalization on classification accuracy is marginal, regardless of its use or non-use. This limited effect may be attributed to the varying characteristics of each dataset, which influence the need for standardization differently. However, in terms of model convergence speed, it's evident that models with normalized preprocessing converge more rapidly. Hence, we recommend the adoption of normalization in the preprocessing stage.
>
> Regarding the suggestion to perform comparative experiments with other time series representation methods in conjunction with the transformer model, we have implemented typical image encoding techniques using both Transformer and CNN architectures for image classification of UTS. The outcomes of these experiments are presented in Figure 3. It's important to highlight that our study is pioneering in applying image representation to Multivariate Time Series (MTS). This approach has demonstrated superiority over other image representation methods and shows competitive performance relative to current state-of-the-art deep learning methods in Time Series Classification (TSC).
>
> Finally, we concur with the assessment of ViT's strengths, particularly its positional embedding and linear projection of flattened image patches. These features were a key factor in selecting ViT for our framework. However, it's crucial to note that ViT alone does not account for our method's effectiveness. As seen in Figure 3, merely utilizing existing image encoding methods followed by ViT yields inferior results compared to baseline models. Regarding Gaussian mapping, we consider it a versatile tool in handling time series data, which often includes noise and fluctuations. This approach can smooth the data, enhancing the robustness of the model.
>
> We hope the above answers can address your questions and concerns.

---

> ### Author Response · Authors · 2023-11-23
> **Official Response to Reviewer mdnq on weakness 2/2**
>
> > W2: lack of motivation for the main method
>
> **Answer**: We thank the reviewer for these insightful comments of our work. The goal of this paper is to propose a novel time series image representation framework that not only has better comprehensive performance compared with existing deep learning SOTA algorithms but also has the inherent generalization ability of both UTS and MTS with inconsistent length. **Inspired by 2D keypoint detection task** in computer vision, we noticed that Gaussian distributions are often used to model uncertainty or confidence in locations. In addition, Gaussian distribution is a smooth distribution; that is to say, since time series data usually have noise and fluctuations, we think modeling it through Gaussian distribution can smooth the data to a certain extent and make the model more robust, which is our motivation for Gaussian mapping. At the same time, the sub-patch constructed from the data of each channel time point can also better integrate the information of each channel, which is helpful for the model to better understand the overall characteristics of multivariable time series.
>
> We also appreciate the reviewer for bringing about the analysis from a data perspective. We did an in-depth analysis based on all the datasets we use. Here are some findings we want to share. First, since the experiment with Time2Image was conducted on an all time series public dataset, the superior performance indicates that this method is suitable for most time series data. Here are some findings we want to share from our perspective:
>
> (1) **The number of channels in MTS will have an important impact on the performance**; that is to say, our method might have limited performance when the number of MTS is large. Since the size of the sub-patch is fixed and we regard data on each dataset from different channels as "equal circle in the square" problem, that means the larger the channel, the smaller the place for representation of each channel (see DuckDuckGeese with 1345 channels, PEMS_SF with 963 channels,etc.).
>
> (2) **The length of the time series will have an impact on the performance**; that is to say, the smaller the difference between predefined sizes L, the better the performance.The reason is mainly because of the resize we use in preprocessing. In this work, we set size L to 196. Even though resizing can deal with unequal length on MTS, it is recommended to find the suitable size based on the original length of the dataset in the future. For instance, when the length of the time series is too long, there will be a large information loss, which might have a negative impact on the performance (see EigenWorms with 17984, CinCECGTorso with 1639, and HandOutlines with 2709).
>
> I hope these findings will address your concern. Any further suggestions and comments are still welcome.

---

> ### Author Response · Authors · 2023-11-23
> **Official Response to Reviewer mdnq on Questions 1/2**
>
> **About Questions**
> > What are the author’s intuitions for why Time2Image underperforms the regular ResNet? Is there something about the Gaussian representation that is particularly suited to Transformers over Convulational approaches?
>
> **Answer**: We appreciate your insightful question regarding the performance comparison between Time2Image and the regular ResNet. As far as we are concerned, the observed difference in performance can be attributed to the unique characteristics of the proposed Gaussian representation ATSGM and its compatibility with Transformers. To illustrate this, let's delve into the Time2Image image representation process using multivariate time series (MTS) as an example. Following the Gaussian representation by ATSGM, each patch encapsulates the image representation of all channels at a specific time stamp from the MTS. This implies that each patch contains local information, particularly channel information, at a certain time stamp. Subsequently, these patches are arranged in chronological order to create input sequences for the Vision Transformer (ViT). This arrangement allows ViT to effectively capture temporal information and learn patterns in a sequential manner. Therefore, our use of Gaussian mapping in ATSGM aligns well with Vision Transformer (ViT) because ViT is excellent at understanding long-term patterns in sequences. The smooth and continuous nature of the Gaussian representation enhances ViT's ability to capture important features in a seamless way, making our approach effective for analyzing complex patterns in time series data.
>
> > The results of InceptionTime seem to worse in this evaluation compared with the results of the Time Series Benchmark and Redux. Is this because T2I is better at the same categories as inception time? While the paper explains the generalization ability of T2I across domains it does not shine any light on where (if any) the gains may come from.
>
> **Answer**: We appreciate your insightful comments and suggestions. From our understanding, the generalization ability of Time2Image might depend on the flexibility of Gaussian representation. The idea of ATSGM is to integrate information from different channels at the same time point, which helps enhance the model's ability to capture diverse features in multivariate time series data. This integration facilitates a more comprehensive understanding of the interdependencies and interactions among different channels. The design choice of combining information from various channels at each time point demonstrates the flexibility of Gaussian representation, allowing it to adapt well to the intricate patterns present in multivariate time series data. This integration aligns with real-world scenarios where inter-channel relationships are crucial, enhancing the model's capacity to accurately represent complex, correlated structures in diverse time series datasets. This can also be proved from our result in Table 6, which we can see that the better performance mainly belongs to the physiological dataset, which has typical characteristics of the correlation and dependence between different channels. We hope it can address your question, and any further comments or thoughts are also welcome to have a better understanding of the framework.
>
> > Are the authors concerned that the method for selecting the standard deviation parameter could allow overfitting on the benchmark datasets? Were there any held-out datasets for parameter selection that could be used for evaluation?
>
> **Answer**:
> Thank you for your questions. In our work, we used pre-defined training and testing splits for fairness comparison to other baselines without having a held-out dataset specifically for parameter selection. However, we would like to note that for the standard deviation parameter selection process, we employed the "3 sigma rule" of the Gaussian distribution. By deriving a relationship between the standard deviation, sub-patch size, and the fixed number of channels in multivariate time series, we can figure out the value of the standard deviation. We really appreciate your insightful question, and we will further elaborate on this aspect in the revised version. Please find the clarification in detail on page 5, which we believe provides additional clarity on the rationale behind the selection of standard deviation parameters, as demonstrated in Table 7.

---

> ### Author Response · Authors · 2023-11-23
> **Official Response to Reviewer mdnq on Questions 2/2**
>
> > Given the restrictions of the algorithm, in that finding a packing for the gaussian distributions in the patches will reduce the resolution of the final sample as the number of channels increases, would it make sense to evaluate the performance of the model as the number of channels increases? From quick observation on the datasets with high number of channels (Crop-7200, phalangesOutlinesCorrect-1800, Ford*-3600, ElectricDevices-8926, NonInvasive*-1800, DuckDuckGeese-1345) it seems that in this collection the T2I win-rate is lower (~50%). A similar analysis would be on sequence length. EigenWorms contains the longest sequences by an order of magnitude in the MTS set and Time2Image performance is much worse than ResNet or InceptionTime. Helping understand these performance bounds would be very useful for practical applications and understanding the generalization from the test benchmark to other evaluations. This could also help inform how the additional parameters for series normalization and patch size can be set.
>
> **Answer**: Thank you for your insightful suggestions and observations. We think this question is similar to the Weakness #2, so please find the answers from weakness #2 to see if those findings can address you questions. We fully agree with your suggestion on understanding the performance bounds under different conditions and how this knowledge can inform the setting of additional parameters. We will elaborate on these aspects in the revised manuscript to provide a more comprehensive understanding of our approach.
>
> > Additional Nits:
> Page 5: the link to the best packings of equal circles in a square returns 404. The correct link should include a ‘#’ before ‘Results’.
> Algorithm 1 does not define L anywhere in the paper though I believe it is simply equal to 196.
> Table 1: Incorrect based on the results of Table 6. In Table 6 ResNet shows 4 wins instead of 3, Time2Image shows 11 wins instead of 13, and Time2Image+ResNet shows 2 wins which should not be conflated with Time2Image+ViT in this table.
> Figure 4 should include InceptionTime
>
> **Answer**: Thank you so much for your careful reading. Based on your suggestions, we made the following revisions to our paper:
> - We revised the link on page 5, and you're right about the correct link
> - We added the definition of L to our revised manuscript on page 4.
> - The Figure 4 is also updated to include the performance of InceptionTime.
> - What we want to note is that after a careful check on Table 1, we think there might be a misunderstanding from the analysis. What we want to show in Table 1 is the performance of our proposed Time2Image compared to other baseline models (ROCKET, CIF, HIVE-COTE, ResNet, and InceptionTime). However, in the last column of Table 6 is the performance of the proposed image representation method for different frameworks of image classification that we want to explore from comparative experiment for further analysis. Therefore, in Table 1, we do not need to consider the performance of the last column from Table 6. Hope this explanation solves your problem.
>
> We're grateful for your thoughtful questions and helpful advice. Please feel free to reach out if you have any additional queries. If you feel that our response adequately addresses your concerns, would you like to consider raising your rating score for our paper? Thank you for your consideration!

---

### Author Response · Authors · 2023-11-23
**General Response**

We would like to express our sincere gratitude to all reviewers for dedicating their valuable time to reviewing our manuscript and providing constructive feedback. We are delighted to see the positive recognition from several reviewers, with three of them acknowledging the novelty of our method. Moreover, we extend our thanks to Reviewer mdnq for noting that our method "provides a reasonable way to represent time series." We also appreciate the commendation from Reviewer HAQG, who found our method "well motivated as a powerful method for computer vision utilization" and recognized its contribution to "advancing the general family of methods". We take every suggestion and comment each reviewer brings up seriously and try to answer it as comprehensively and clearly as possible, followed by some additional experiments if needed. Please find the answers to each comment.

We also noticed that more than one reviewer showed great interest in our motivation and intuition for the work. In this response, we would like to address this by providing detailed explanations of the motivation, advantages, and contributions of our work to ensure a better understanding and positioning within the field. Please find them in the following:

We are working on time series classification through image representation, which is a small but challenging field. There are 2 **research gaps** in this field: (1) Before this work, there was no research on multivariate time series (MTS) image representations since it was a challenging task. (2) Even though there are some image representation approaches for univariate time series, there are significant performance gaps to make a comparison of current state-of-the-art (SOTA) models in classification tasks. Therefore, our **goal** is to propose a novel time series image representation framework that not only has better comprehensive performance compared with existing deep learning SOTA algorithms but also has the inherent generalization ability to both UTS and MTS with inconsistent length.Inspired by **the 2D keypoint detection** task in computer vision, we noticed that Gaussian distributions are often used to model uncertainty or confidence in locations. In addition, a Gaussian distribution is a smooth distribution; that is to say, since time series data usually have noise and fluctuations, we think modeling it through a Gaussian distribution can smooth the data to a certain extent and make the model more robust. Therefore, introducing this idea into the representation of time series data by modeling the uncertainty of each channel time point and forming a sub-patch through the Adaptive Time Series Gaussian Mapping (ATSGM) we proposed can help the model better understand the importance of each time point in the time series data. At the same time, the relationship and local information between different channels in MTS data can also be obtained in order to improve the model's robustness and generalization ability. Therefore, in our framework, we hope the time series image representation can better capture local temporal and channel information, while the transformer structure ViT adopts can effectively capture long-term information. This is our original idea for ​​designing this method.

This work is the **first work** to propose a time series image representation suitable for both UTS and MTS with unequal length, and the ViT-B/16 we adopted is also the **first attempt** on time series classification. In addition to that, experiments were conducted on all 158 public time series datasets from UCR/UEA covering all UTS and MTS, and shows superiority of the algorithm by comparing deep learning SOTA algorithms. We hope the above description will give you more information about the motivation and intuition of this framework.

In addition, we have addressed questions and comments in our responses to each reviewer and revised our paper. Key updates in the revised paper included: (1) Additional experiment results were added to better position our work, which can be seen in the Appendix. (2) Additional information is added to Figures 1 and 2 for a better understanding of the framework. (3) A detailed analysis of the performance based on datasets of different lengths, domains, etc to find the bounds of the work.

Thank you again for your time and engagement. If you have any more questions, we are happy to continue the discussion even after the discussion process. We are also willing to address the problems with the public if possible.

---

### Meta-Review · Area_Chair_Xs9S · 2023-12-10

**Metareview:**

This paper presents an image-based representation learning framework called Time2Image for time series classification. In particular, the proposed Adaptive Time Series Gaussian Mapping (ATSGM) module works well for the time-series to image conversion. Overall, I would think this paper has certain merits but it falls below the ICLR standard. I agree on the concerns raised by the reviewers, particularly the concerns on literature review and experimental evaluation. For example, there are several language-based (OneFitsAll) and other image-based (TimesNet and PIP) frameworks for time series analysis, while TimesNet outperforms Time2Image as reported in the rebuttal. The effectiveness of the proposed method currently lacks sufficient support. Therefore, my recommendation is to reject this paper.

**Justification For Why Not Higher Score:**

The effectiveness of the proposed method currently lacks sufficient support, due to inadequate literature review and unconvincing evaluation. Therefore, we are not able to accept this paper.

**Justification For Why Not Lower Score:**

N/A.

---

### Decision · Program_Chairs · 2024-01-16

Reject